# An increase in methane emissions from tropical Africa between 2010 and 2016 inferred from satellite data

Mark F. Lunt[1], Paul I. Palmer[1,2], Liang Feng[1,2], Christopher M. Taylor[3,4], Hartmut Boesch[5,6], and Robert J. Parker[5,6]

[1]School of GeoSciences, University of Edinburgh, Edinburgh, UK
[2]National Centre for Earth Observation, University of Edinburgh, Edinburgh, UK
[3]Centre for Ecology and Hydrology, Wallingford, UK
[4]National Centre for Earth Observation, Wallingford, UK
[5]Earth Observation Science, Department of Physics and Astronomy, University of Leicester, Leicester, UK
[6]National Centre for Earth Observation, University of Leicester, Leicester, UK

**Correspondence:** Mark Lunt (mark.lunt@ed.ac.uk)

**Abstract.**

Emissions of methane ($CH_4$) from tropical ecosystems, and how they respond to changes in climate, represent one of the biggest uncertainties associated with the global $CH_4$ budget. Historically, this has been due to the dearth of pan-tropical *in situ* measurements, which is particularly acute in Africa. By virtue of their superior spatial coverage, satellite observations of atmospheric $CH_4$ columns can help to narrow down some of the uncertainties in the tropical $CH_4$ emission budget. We use proxy column retrievals of atmospheric $CH_4$ ($XCH_4$) from the Japanese Greenhouse gases Observing SATellite (GOSAT) and the nested version of the GEOS-Chem atmospheric chemistry and transport model ($0.5° \times 0.625°$) to infer emissions from tropical Africa between 2010 and 2016. Proxy retrievals of $XCH_4$ are less sensitive to scattering due to clouds and aerosol than full physics retrievals but the method assumes that the global distribution of carbon dioxide ($CO_2$) is known. We explore the sensitivity of inferred *a posteriori* emissions to this source of systematic error by using two different $XCH_4$ data products that are determined using different model $CO_2$ fields. We infer monthly emissions from GOSAT $XCH_4$ data using a hierarchical Bayesian framework, allowing us to report seasonal cycles and trends in annual mean values. We find mean tropical African emissions between 2010–2016 range from 76 (74–78) Tg yr$^{-1}$ to 80 (78–82) Tg yr$^{-1}$, dependent on the proxy $XCH_4$ data used, with larger differences in northern hemisphere Africa than southern hemisphere Africa. We find a robust positive linear trend in tropical African $CH_4$ emissions for our seven-year study period, with values of 1.5 (1.1–1.9) Tg yr$^{-1}$ or 2.1 (1.7–2.5) Tg yr$^{-1}$, dependent on the $CO_2$ data product used in the proxy retrieval. This linear emissions trend accounts for around a third of the global emissions growth rate during this period. A substantial portion of this increase is due to a short-term increase in emissions of 3 Tg yr$^{-1}$ between 2011 and 2015 from the Sudd in South Sudan. Using satellite land surface temperature anomalies and altimetry data we find this increase in $CH_4$ emissions is consistent with an increase in wetland extent due to increased inflow from the White Nile, although the data indicate that the Sudd was anomalously dry at the start of our inversion period. We find a strong seasonality in emissions across northern hemisphere Africa, with the timing of the seasonal emissions peak coincident with the seasonal peak in ground water storage. In contrast, we find that *a posteriori* $CH_4$ emissions from the

wetland area of the Congo basin are approximately constant throughout the year, consistent with less temporal variability in wetland extent, and significantly smaller than *a priori* estimates.

## 1 Introduction

The recent and ongoing rise in atmospheric $CH_4$ since 2007, after a period of relative stability, has been well documented, although the causes are still not fully understood (e.g. Rigby et al., 2008; Nisbet et al., 2014; Turner et al., 2019). Dominant sources of $CH_4$ to the atmosphere are both natural and anthropogenic including fossil fuels, agriculture, waste management and natural wetlands (Kirschke et al., 2013; Saunois et al., 2016), whilst the major sink is due to reaction with the hydroxyl radical (OH) in the troposphere. Several hypotheses have been suggested that could explain recent changes in atmospheric $CH_4$

but none are verifiable because of a lack of data at the global scale (Turner et al., 2019). These hypotheses include increased fossil fuel emissions, increased microbial emissions, or some combination of the two allied with other factors (e.g. Schaefer et al., 2016; Hausmann et al., 2016; Worden et al., 2017; McNorton et al., 2018; Thompson et al., 2018). Additionally, we cannot discount a role for a changing OH sink based on $CH_4$, isotopic $\delta^{13}CH_4$ and methyl chloroform observations (Rigby et al., 2017; Turner et al., 2017).

One of the plausible explanations is that tropical microbial emissions have increased (Nisbet et al., 2016; Schaefer et al., 2016; Thompson et al., 2018). This hypothesis is based largely on a significant negative trend in $\delta^{13}CH_4$ isotope values globally and the latitudinal distribution of $CH_4$ growth rates. Microbial sources are more depleted in $\delta^{13}CH_4$ than other sources so that a move to lighter isotope values can be interpreted as microbial sources providing a greater proportion of total $CH_4$ emissions. However, $\delta^{13}CH_4$ source signatures for different microbial sources and their variation over time are not well characterized

(Turner et al., 2019). As a result the constraint provided by isotope data is limited to broad-scale inferences on changes in sources and sinks, and cannot narrow down which type of microbial source is responsible. Additional independent information, such as that from process-based based wetland models, can provide further evidence for changes in microbial sources. For example, some individual wetland model studies suggest that wetland $CH_4$ emissions have increased (e.g. McNorton et al., 2016; Zhang et al., 2018), although the increases are relatively small and likely to be model dependent (Poulter et al., 2017).

One of the main challenges associated with studying wetland emissions of $CH_4$ is that they are difficult to describe mechanistically. Process-based wetland models use parameterizations of biological processes informed by field data, together with estimates of the spatial extent of wetland, to describe the seasonal magnitude and distribution of wetland emissions across the globe. The extent of wetland area is usually prescribed from climatology (Lehner and Doll, 2004; Bergamaschi et al., 2007), determined from a hydrological model (Gedney and Cox, 2003) or parameterized using remotely sensed inundation datasets

(Prigent et al., 2007; Schroeder et al., 2015). Because the spatiotemporal variability of wetland extent is key to estimating $CH_4$

emissions, disagreement between different measures of wetland areas equates to a wide range of emission estimates (Melton et al., 2013).

The African continent contains significant microbial methane sources from wetlands and agricultural enteric fermentation, as well as smaller microbial sources from termites and wild ruminants (Crutzen et al., 1986; Sanderson, 1996). A recent study, comprising an ensemble of wetland emissions models, estimates African wetland $CH_4$ emissions represent 12 (7–23)% of global wetland emissions (Bloom et al., 2017), where the numbers in parentheses indicate the 95 percentile range. These emissions are concentrated in the sub-Saharan tropics, where we focus our work. Figure 1(a) shows our study domain that includes all the main wetland regions within Africa, including the Congo Basin located in Central Africa and the Sudd in South Sudan. For the purposes of our study we use a very broad definition of wetlands to include any areas of land that are permanently or periodically inundated.

Due to the seasonal migration of the Intertropical Convergence Zone (ITCZ), rains fall in the sub-Saharan lands of northern hemisphere Africa during boreal summer, and in the southern hemisphere Africa during austral summer. Wetland extents vary with this delineation of wet and dry seasons, with maximum wetland extents generally occurring at the end of each rainy season (Taylor et al., 2018). The contribution from different wetlands to continental-scale $CH_4$ emissions is uncertain. Consequently, there is considerable disagreement between wetland emission models about the distribution and magnitude of $CH_4$ emissions, particularly regarding the timing of the seasonal peak of emissions between 0–15 °N, with models predicting peak emissions to occur in February, April–October or September–November (Bloom et al., 2017). The uncertainty in African $CH_4$ emissions is compounded by uncertainties associated with emissions from seasonal fires, and agricultural $CH_4$ emissions, especially enteric fermentation from livestock. Enteric fermentation estimates are based on uncertain activity data and Tier 1 emissions factors from the Intergovernmental Panel on Climate Change (IPCC), with studies suggesting these emission factors are either too low (Kouazounde et al., 2014; Toit et al., 2014; Wolf et al., 2017) or too high (Goopy et al., 2018), and should be country dependent.

The tropics, particularly Africa, are generally poorly sampled by *in situ* atmospheric measurement networks. As such, this is where satellite data have the greatest potential to develop current understanding of $CH_4$ emissions, despite the requirements for cloud-free scenes. Bloom et al. (2010, 2012) used dry air column averaged methane mole fraction ($XCH_4$) data from the SCIAMACHY satellite (Frankenberg et al., 2011) and Gravity Recovery And Climate Experiment (GRACE, (Tapley et al., 2004)) liquid water equivalent (LWE) height anomaly retrievals to show that the seasonal cycle of wetland $CH_4$ emissions can largely be explained by seasonal changes in water volume in the tropics, and temperature in the high latitudes. More recently, Parker et al. (2018) compared peak-to-peak seasonal cycles of $XCH_4$ from models and the Greenhouse gases Observing SATellite (GOSAT) over tropical wetlands including those in Africa. They find significant discrepancies between model estimates, driven by a range of wetland emission estimates, but do not determine the magnitude of the $CH_4$ emissions that are consistent with the GOSAT data.

In this work, we infer $CH_4$ emissions over tropical Africa from 2010 to 2016 (Figure 1) from GOSAT $XCH_4$ data using the GEOS-Chem 3-D chemistry transport model, driven by *a priori* emission estimates, and a Bayesian inversion method. The data and methods we use are described in section 2. Previous studies of atmospheric $CH_4$ using satellite data tend to be global in

scope with *a posteriori* emission estimates inferred over large continental regions (e.g. Fraser et al., 2013; Pandey et al., 2016; Feng et al., 2017), although some studies use satellite data in regional inversions to infer emissions on smaller regional scales (Turner et al., 2015; Ganesan et al., 2017; Miller et al., 2019). Recent work using GOSAT XCH$_4$ data has suggested that both atmospheric CH$_4$ mole fractions and CH$_4$ emissions from the African continent have increased since 2009 (Maasakkers et al., 2019; Miller et al., 2019). Here, we take advantage of a nested capability of GEOS-Chem, allowing us to study atmospheric CH$_4$ with a spatial resolution of $0.5° \times 0.625°$, driven by coarser resolution lateral boundary conditions provided by the same model. We report our results in section 3, including regional seasonal and annual mean variations. We focus our attention on significant increased emissions from the Sudd wetlands and use correlative data to propose the underlying mechanism. We conclude the paper in section 4.

## 2    Data and Methods

### 2.1    GOSAT XCH$_4$ data

GOSAT was launched in 2009 into a sun-synchronous orbit with a local equator crossing time of 13:00, resulting in global coverage every three days (Kuze et al., 2009). We use data from the Thermal And Near infrared Sensor for carbon Observations - Fourier Transform Spectrometer (TANSO-FTS) that measures short wave infrared (SWIR) radiances between 0.76–2.0 $\mu$m at a resolution of 0.3 cm$^{-1}$ from which dry air column averaged CH$_4$ values are retrieved (Parker et al., 2011). We use the University of Leicester's (UoL) v7 GOSAT-OCPR proxy XCH$_4$ product (Parker et al., 2015). These data have been previously validated against ground-based data from the Total Carbon Column Observing Network (TCCON, Wunch et al., 2011), with a global mean bias of +4.8 ppb and a single sounding precision of 13.4 ppb (Parker et al., 2015). The TCCON data available in the tropics agree reasonably well with these GOSAT data, with a positive bias of 3.1 ppb at Ascension Island and 3.4 ppb at Paramaribo, Suriname (Parker et al., 2018). However, there are currently no TCCON stations in tropical Africa.

We use GOSAT-OCPR proxy XCH$_4$ data over our study domain described by 20°W–55°E, 26°S–26°N. Retrievals are filtered for cloud-free scenes by retaining data where the surface pressure difference between the O$_2$-A band apparent surface pressure retrieval and the corresponding ECMWF surface pressure is within 30 hPa and where the signal-to-noise ratio is above 50 (Parker et al., 2015). We use both nadir and glint observations in this work, although we assess the impact of using only nadir observations to infer *a posteriori* CH$_4$ emissions in Section 3.1.

The proxy XCH$_4$ data is given by:

$$\mathrm{XCH_{4\,proxy}} = \frac{\mathrm{XCH_4}}{\mathrm{XCO_2}} \times \mathrm{XCO_{2\,model}}, \tag{1}$$

where XCH$_4$ and XCO$_2$ are the retrieved column values of the CH$_4$ and CO$_2$ respectively (Parker et al., 2015). These values are retrieved in neighbouring spectral windows at 1.65 $\mu$m and 1.61 $\mu$m for CH$_4$ and CO$_2$ respectively, meaning that common factors that impact the retrievals, e.g. aerosol and cloud scattering, can be removed by taking the ratio. The "proxy" column CH$_4$ is then determined by multiplying by an independent estimate of the XCO$_2$, typically from an atmospheric model. In using the ratio to determine XCH$_4$, we assume that atmospheric CO$_2$ varies much less than CH$_4$. The main advantage of this

data product is that it is more robust than the full physics approach in the presence of clouds and aerosols. This is particularly important over the tropics where clouds are prevalent during the wet season and biomass burning aerosols are widespread during the dry season. However, this approach propagates model $CO_2$ errors to the proxy $XCH_4$ data product and subsequently to *a posteriori* emission estimates.

Here, we assess the sensitivity of our *a posteriori* $CH_4$ emission estimates to different model fields of $CO_2$. Our control $XCH_4$ data, PR1, is based on the standard UoL $XCH_4$ dataset, which uses the median $XCO_2$ field from three models (Parker et al., 2015). The second $XCH_4$ data product, PR2, uses $XCO_{2\ model}$ generated using a consistent version of the GEOS-Chem atmospheric chemistry transport model (described below) run at a spatial resolution of $4° \times 5°$ with emissions inferred using an ensemble Kalman filter (Feng et al., 2016) from the GOSAT ACOS $XCO_2$ data (O'Dell et al., 2012). Previous work has
shown that $CO_2$ flux estimates from GOSAT for northern tropical Africa are larger than those inferred from *in situ* $CO_2$ mole fraction data (Houweling et al., 2015). Consequently, PR2 $XCH_4$ proxy data are significantly different from PR1 $XCH_4$ data.

Figure 2 shows maps of the annual mean difference between the PR1 and PR2 $XCH_4$ data from 2010 to 2016. In general, the PR2 $XCH_4$ data are larger than the PR1 $XCH_4$, with a mean difference of 1.6 ppb over the seven-year period. The largest differences ($\simeq 10$ ppb) are found over West Africa in 2010, over the Democratic Republic of Congo and Southern Africa in
2015 and over almost all of tropical Africa in 2016. In 2016, the mean of the PR2 $XCH_4$ data was 3.1 ppb larger than PR1. The larger PR2 $XCH_4$ levels in 2016 result from large net $CO_2$ emissions over tropical Africa inferred from satellite observations of $XCO_2$ (Palmer et al., 2019). With the exception of 2016, the smallest annual mean differences are generally observed over East Africa and north of 15°N. The spatial heterogeneity of the differences shown in Figure 2 illustrates the importance of assessing the sensitivity of the inversions to different $XCH_4$ proxy datasets, and we present inversion estimates based on both
sets of data in our results.

## 2.2  GEOS-Chem Atmospheric Chemistry Transport Model

We use a nested version of the GEOS-Chem atmospheric chemistry transport model (v11-01, Bey et al., 2001; Wecht et al., 2014; Turner et al., 2015) to relate surface emissions of $CH_4$ to atmospheric mole fractions. The model is driven by MERRA-2 reanalysis meteorological fields (Bosilovich et al., 2016), provided by the Global Modeling and Assimilation Office (GMAO)
at NASA Goddard Space Flight Center. The native spatial resolution of these data is 0.5°(latitude)×0.625° (longitude), and includes 47 vertical terrain-following sigma-levels that describe the atmosphere from the surface to 0.01 hPa of which about 30 are typically below the dynamic tropopause. The 3-D meteorological data are updated hourly, and 2-D fields and surface fields are updated every 3 h. The atmospheric transport and chemistry time steps are five and ten minutes, respectively.

The nested region is focused on tropical Africa (Figure 1) described by 20°W–55°E, 26°S–26°N. Input emission fields
to the nested GEOS-Chem run included wetlands (WetCHARTS, Bloom et al., 2017), biomass burning (GFED, v4, van der Werf et al., 2017), anthropogenic sources including agriculture, waste and energy (EDGAR, v4.3.2, Janssens-Maenhout et al., 2017), and termites (Fung et al., 1991). Both WetCHARTs and GFED *a priori* emissions vary monthly between 2010–2016. EDGAR emissions from 2012 were used for all years for anthropogenic emissions. $CH_4$ loss due to oxidation by the hydroxyl radical (OH) in the troposphere is computed using a climatology of monthly OH concentration fields from a full-chemistry

GEOS-Chem simulation, resulting in a lifetime with respect to tropospheric OH of 9.9 years (Wecht et al., 2014). Additional minor sinks of stratospheric oxidation (Murray et al., 2012) and a soil sink term (Fung et al., 1991) result in a global mean $CH_4$ atmospheric lifetime of 8.8 years.

At the boundaries of the nested domain we use time-dependent lateral boundary condition fields from a global optimization of $CH_4$ fluxes, described in Feng et al. (2017), the details of which we briefly summarise here. Global monthly methane emissions were optimized from spatial regions that were sub-divisions of the TransCom-3 regions using an ensemble Kalman filter approach from *in situ* $CH_4$ data. The global GEOS-Chem model at $4° × 5°$ was used to relate fluxes to the observations from the National Oceanic and Atmospheric Administration (NOAA) network of sites. The resulting *a posteriori* fluxes were used to generate the global $CH_4$ fields for use in the lateral boundary conditions of the nested model, with the forward model initialised in January 2009 to minimize the influence of initial conditions. We stress that these global model boundary condition fields were only used *a priori* in our nested model inversions, and updated in our inversion via *a posteriori* scaling factors based on the GOSAT data.

Previous work using GEOS-Chem in GOSAT $XCH_4$ inversion studies have included a latitudinally dependent bias term, which has been attributed to errors in the stratospheric transport of $CH_4$ within the GEOS-Chem model (Turner et al., 2015; Maasakkers et al., 2019). The form of this model bias is most pronounced at high latitudes and is particularly prevalent in the $4x5°$ model, and thus may be resolution dependent (Maasakkers et al., 2019). Given our region of study is in the tropics where the bias is small and we use a much higher resolution model we have not included a bias correction term in this work. However, we include some sensitivity tests to this potential model bias in Section 3.4.

### 2.3 Hierarchical Bayesian Inversion Method

To infer *a posteriori* $CH_4$ fluxes from the GOSAT $XCH_4$ data between January 2010 to December 2016 we use a hierarchical Bayesian inversion method. We refer the reader to Ganesan et al. (2014) and Lunt et al. (2016) for full details of the method as applied to atmospheric inversions. Here, we provide a brief overview of the method and the outline the specifics of the set-up as applied to this work. We solve for a state vector of emissions, $\mathbf{x}$, which are related to the data, $\mathbf{y}$, through the forward model in equation 2:

$$\mathbf{y} = \mathbf{H} \cdot \mathbf{x} + e, \tag{2}$$

where $\mathbf{H}$ is the sensitivity matrix, describing the response of atmospheric columns of $CH_4$ to changes in $\mathbf{x}$. The term $e$ accounts for random errors in both model and measurements in the forward model. In addition to solving for $\mathbf{x}$, the hierarchical Bayesian approach also takes into account uncertainties in the parameters, $\boldsymbol{\theta}$ that govern the form of the probability density function (PDF) of $\mathbf{x}$, such as the standard deviation for the case of a Gaussian PDF. The full form of the hierarchical Bayesian equation is given by:

$$\rho(\mathbf{x}, \boldsymbol{\theta}|\mathbf{y}) \propto \rho(\mathbf{y}|\mathbf{x}, \boldsymbol{\theta}) \cdot \rho(\mathbf{x}|\boldsymbol{\theta}) \cdot \rho(\boldsymbol{\theta}). \tag{3}$$

The $\rho$ terms are used to denote a PDF. Equation 3 indicates that uncertainties in $\boldsymbol{\theta}$ propagate through to the *a posteriori* distribution of $\mathbf{x}$ on the left hand side of equation 3, which is one of the benefits of this approach given the *a priori* and model

uncertainties may not be well known. In addition, the choice of *a priori* or measurement PDF is open to the investigator, which allows flexibility in cases where, for example, the PDF should be defined only on the positive axis. Since it cannot be solved analytically we estimate the *a posteriori* distribution of $\mathbf{x}$ using a Markov chain Monte Carlo (MCMC) approach.

At each step in the Markov chain a new state (e.g., new emissions value) is proposed, based on the current value of the state. Whether that proposal is accepted or not depends on the following criterion:

$$U \leq \left( \frac{\rho(\mathbf{x}')}{\rho(\mathbf{x})} \times \frac{\rho(\mathbf{y}'|\mathbf{x}')}{\rho(\mathbf{y}|\mathbf{x})} \right), \tag{4}$$

where $U$ is a uniformly-distributed random number between zero and one, and $\rho$ represents a PDF. The term $\rho(\mathbf{x})$ represents the *a priori* PDF of the current state and $\rho(\mathbf{x}')$ the new proposed state. $\rho(\mathbf{y}|\mathbf{x})$ and $\rho(\mathbf{y}'|\mathbf{x}')$ represent the likelihood functions of the current and proposed states respectively. Consequently, the probability of a new proposal being accepted depends on the balance of the ratio between proposed and current *a priori* PDFs (how far the new state is from the *a priori*) and the likelihood ratio (how well it fits to the data). This procedure is repeated many thousands of times to build up an *a posteriori* PDF.

The state vector in this work is composed of terms representing scale factors applied to emissions, boundary conditions and initial conditions. We calculate the monthly emissions from a total of 38 regions across tropical Africa (Figure 1(b)). These regions are informed by national boundaries and major river basins or wetlands such as the Congo and Sudd. We include an additional six basis functions to describe monthly boundary condition scaling to the global $CH_4$ model field at the north, north-east, south-east, south, south-west and north-west boundaries. We also include a basis function to describe the initial 3-D $CH_4$ mole fraction distribution within the nested domain at the start of the inversion period. In total, we calculated 3697 basis functions over our seven-year study period. We describe *a priori* distributions for each emissions basis function as a lognormal PDF, with an arithmetic standard deviation of 100%, reflecting the large uncertainty in the *a priori* estimates. *A priori* uncertainties on the boundary condition scaling factors were 0.5%, equivalent to a 10 ppb uncertainty on the mean value of each monthly field.

We use the uncertain parameters vector $\boldsymbol{\theta}$, to describe the model-measurement uncertainty that governs the form of the likelihood PDF, $\rho(\mathbf{y}|\mathbf{x}, \boldsymbol{\theta})$, in equation 3. Model-measurement uncertainties were defined as a combination of the measurement uncertainties (which were fixed in the inversion) and a model uncertainty represented by the $\boldsymbol{\theta}$ parameters vector. The measurement uncertainty term represents the statistical uncertainties in the retrieved $XCH_4$ data, and averaged 9 ppb across all data points between 2010–2016 (Parker et al., 2015). The same measurement uncertainty values are used for both PR1 and PR2 data. The model uncertainty term was defined by a uniform distribution with lower and upper bounds of 5 ppb and 25 ppb respectively, with separate values estimated for model uncertainties in $8° \times 10°$ bins, and every month.

Due to the density of data, and to make the inversions more computationally tractable we assumed measurement errors were spatially and temporally uncorrelated. Although this might result in an underestimate of the posterior uncertainties, one approach to form a diagonal approximation of a full covariance matrix is through error inflation (Chevallier, 2007). The inclusion of a model uncertainty term in our inversions, in addition to the measurement uncertainty, partly acts to inflate the error. In our inversions this term is allowed to vary, and constrained by the data, but is of a similar magnitude to the measurement uncertainty, with a mean a posteriori value of 8 ppb, leading to an increase in the error term. In addition, Turner et al. (2015)

suggested that the measurement uncertainties of Parker et al. (2011) were already a conservative estimate of uncertainty, further adding to the observation error inflation.

The sensitivity matrix $\mathbf{H}$ was calculated by running the forward model for one month, with emissions in each basis function based on the *a priori* distribution. After one month the emissions were turned off and we tracked the 3-D $CH_4$ mole fraction field for a further three months as the signal decayed in the study domain due to atmospheric transport processes. After three months the residual signal due to perturbed emissions was negligible, so that we assumed zero sensitivity after this point. We sampled the model at the time and location of each GOSAT observed scene, and convolved the resulting model $CH_4$ profile with scene-dependent GOSAT averaging kernels to create the sensitivity matrix, $\mathbf{H}$.

In our inversions, we ran the MCMC for 25000 iterations, the first 5000 of which were discarded as a "burn-in" period. The resulting chain was thinned every 10th iteration to reduce storage, so that the *a posteriori* distribution of each element of the state vector contained 2000 samples. Automatic tuning of proposal step sizes was performed during the burn-in period to ensure efficient ratios of acceptance between 20–50% (Tarantola, 2005). Updates were proposed to all elements of $\mathbf{x}$ every second iteration. At iterations in between, updates were proposed to all the $\boldsymbol{\theta}$ values.

## 3   Results

We find the 2010–2016 mean *a posteriori* $CH_4$ emission estimate from tropical Africa is 76 (74–78) Tg yr$^{-1}$ from the PR1 $XCH_4$ inversion, and 80 (78–82) Tg yr$^{-1}$ from PR2 $XCH_4$. The numbers reported in brackets represent the 95% uncertainty range between the 2.5th and 97.5th percentiles of the *a posteriori* distribution. The difference between the estimates inferred from the PR1 and PR2 $XCH_4$, without any overlap considering the uncertainty range, clearly shows how different assumptions in $XCO_2$ (equation 1) can impact the corresponding $CH_4$ emission estimates. The difference between PR1 and PR2 also indicates that the inability to account for these systematic uncertainties in the inversions leads to an underestimate of *a posteriori* uncertainty when using a single data product.

The difference between PR1 and PR2 inversions was largely due to differences in northern hemisphere tropical Africa (NH Africa) emissions. Mean PR2 emission estimates from each of the East, West and Central Africa regions, which are all largely or entirely contained in NH Africa, were each 2 Tg yr$^{-1}$ larger than PR1 inversions. The mean NH Africa emissions from PR1 was 44 (42–45) Tg yr$^{-1}$, compared to 48 (47–50) Tg yr$^{-1}$ from PR2. This is directly related to the differences between PR1 and PR2 $XCH_4$ data (Figure 2). The differences between our PR1 and PR2 emissions were greatest in 2010 due largely to West Africa emissions differences, consistent with the differences due to assumptions about $CO_{2\ model}$ shown in Figure 2. In contrast, mean *a posteriori* emissions from Southern Hemisphere Africa (SH Africa) between 2010 and 2016 were very similar with both PR1 and PR2 estimates of 32 (30–34) Tg yr$^{-1}$, reflecting a greater consistency of the underlying $CO_{2\ model}$ fields and hence proxy $XCH_4$ in SH Africa.

Our continental-scale results are consistent with previous estimates of African emissions from global $CH_4$ inversions. Saunois et al. (2016) report total African emissions in 2012, determined from a mean of 25 inversions, of 85 (70–106) Tg yr$^{-1}$. Our *a posteriori* estimates for 2012 were 73 (71–75) Tg yr$^{-1}$ for PR1 and 77 (75–79) Tg yr$^{-1}$ for PR2, although our study

domain (Figure 1) does not include the extreme northern and southern regions of Africa. If we extrapolate our results to the African continent, based on the ratio of our *a priori* tropical African emissions to total continental Africa emissions of 90%, we estimate total African emissions of 81 (79–84) Tg yr$^{-1}$ from PR1 and 85 (83–88) Tg yr$^{-1}$ from PR2, well within the range of Saunois et al. (2016).

Figure 3 shows the 2010–2016 mean spatial distribution of our *a posteriori* CH$_4$ emissions estimates from PR1, the corresponding mean *a priori* emission distribution, and the difference between the two. Although emissions are shown at grid scale resolution of $0.5° \times 0.625°$, emissions were resolved using the coarser basis functions shown in Fig. 1(b), and the *a posteriori* fluxes are representative of these coarse scalings to the *a priori* grid scale distributions. We find consistent differences between the *a posteriori* distribution and *a priori* from certain areas in both PR1 (Figure 3) and PR2 inversions (not shown). For ex-

ample, *a posteriori* emissions from East Africa (31 (30–33) Tg yr$^{-1}$ for PR1 and 33 (31–34) Tg yr$^{-1}$ for PR2) were almost twice as large as *a priori* emissions (17 Tg yr$^{-1}$) over the same region. East African *a posteriori* mean emissions account for 40% of the mean tropical African emissions in both PR1 and PR2 inversions. Our southern Africa *a posteriori* estimates (12 (11–14) Tg yr$^{-1}$ for PR1 and PR2 inversions) were also larger than *a priori* emissions (7 Tg yr$^{-1}$).

This is further highlighted in Fig. 4, which shows the difference between the multi-year average *a posteriori* distributions

from both PR1 and PR2 inversions and the *a priori* mean for each of the basis function regions. The *a priori* mean value lies outside the 95 percentile range for many of the distributions over the East African regions. In contrast, as shown in Fig. 3 and Fig. 4 we find significantly smaller 2010-2016 mean emissions over the Congo basin in Central Africa, particularly the central region of the basin known as the Cuvette Centrale. *A posteriori* emission estimates for PR1 (3.3 (2.7–4.0) Tg yr$^{-1}$) and PR2 (3.4 (2.7–4.1) Tg yr$^{-1}$) were very similar to each other and much smaller than the *a priori* emission estimate (8.5 Tg yr$^{-1}$).

Wetland emissions accounted for more than 90% of *a priori* Congo Basin emissions, implying that the WetCHARTS ensemble mean used in the prior is an over-estimate of the wetland component of emissions in the Congo Basin. The WetCHARTS database provides a full multi-model ensemble of 324 models which have a large range of 2–21 Tg yr$^{-1}$ for annual mean emissions from the Congo Basin and our *a posteriori* mean emissions for the region are more consistent with the models at the lower end of this range. This general finding of smaller *a posteriori* emissions than WetCHARTS in the Congo is consistent

with the spatial differences shown in Maasakkers et al. (2019). Our *a posteriori* estimate is more consistent with Tathy et al. (1992), who estimated methane emissions of 1.6–3.2 Tg yr$^{-1}$ from the flooded forest zone of the Congo basin, based on static chamber measurements of methane flux.

### 3.1 Tropical African CH$_4$ Emission Trends

Figure 5(a) shows our *a posteriori* annual emission estimates from the tropical African region between 26°S–26°N. The results

from the PR1 inversion indicate tropical African emissions increased from 72 (70–73) Tg in 2010 to 84 (82–86) Tg in 2016, with a mean positive trend in monthly emissions of +2.1 (1.7–2.5) Tg yr$^{-1}$. The PR2 estimates show a smaller difference between 2010 and 2016, at 78 (76–80) Tg yr$^{-1}$ and 84 (82–86) Tg yr$^{-1}$ respectively, although there was a substantial dip in 2011 emissions. Consequently, the mean trend in monthly emissions for PR2 was smaller than PR1 but still positive, +1.5 (1.1–

1.9) Tg yr$^{-1}$. This suggests that the overall pattern of increasing CH$_4$ emissions from tropical Africa is robust to assumptions about CO$_2$ made in the XCH$_4$ data.

Figure 5(b) shows the increase in tropical African emissions is driven mostly by emissions from East Africa in both inversions. Monthly *a posteriori* emissions from East Africa increased at a rate of +1.6 (1.3–1.9) Tg yr$^{-1}$ in PR1 and +1.3 (1.0–1.6) Tg yr$^{-1}$ in PR2, where the mean trend represents 80–90% of the total tropical Africa increase from the respective inversions. We found no consistent regional trends from the other African region between the PR1 and PR2 inversions over the inversion period, although there was substantial inter-annual variability which might mask any underlying trends particularly over West Africa, and both inversions show an increase of 4 (2–7) Tg yr$^{-1}$ in Southern African emissions between 2014 and 2016.

We investigated the added benefit of including glint measurements by running a sensitivity calculation using only high-gain nadir measurements, which account for 75% of all observations. We designate these inversions PR1$_{nadir}$ and PR2$_{nadir}$. Mean emissions for the PR1$_{nadir}$ inversion were 75 (72–78) Tg yr$^{-1}$ compared to 76 (74–78) Tg yr$^{-1}$ from PR1, and the PR2$_{nadir}$ mean emissions were 80 (78–83) Tg yr$^{-1}$ compared to 80 (78–82) Tg yr$^{-1}$ from PR2. Therefore, when averaged over multiple years the differences were negligible. However, we find that omitting the glint data impacted the magnitude of the derived emission trends, with mean values 0.5–0.7 Tg yr$^{-1}$ smaller than those inferred with both types of data at 1.4 (0.9–1.9) Tg yr$^{-1}$ and 1.0 (0.5–1.5) Tg yr$^{-1}$ for PR1$_{nadir}$ and PR2$_{nadir}$ respectively. This difference is due to greater constraint on the boundary condition terms in the inversions when using the glint observations (which are more sensitive to the scale factors applied to the domain boundaries due to their proximity to the edge of the domain). Whilst the trends in tropical African emissions from the nadir-only inversions between 2010–2016 are smaller, they remain positive and overlap within the uncertainty bounds with the PR1 and PR2 results. In common with the findings from PR1 and PR2 we find that 80–90% of the mean emissions trend can be explained by the mean trend in emissions from East Africa in the PR1$_{nadir}$ and PR2$_{nadir}$ inversions respectively.

To understand why East African CH$_4$ emissions play such an important role in the tropical African budget, we now examine the *a posteriori* emissions from more localized regions. Over East Africa, we find positive emission trends over a number of basis functions. However, given inter-annual variability and the *a posteriori* uncertainties in monthly emissions, the only basis function region that had a linear trend of monthly emissions with a p value less than 0.05 in both PR1 and PR2 inversions was in South Sudan, where the Sudd wetlands are located. For South Sudan, we find a linear trend of +0.4 Tg yr$^{-1}$ (p=0.01) in both PR1 and PR2 inversions from this region. In the following section we present evidence that supports that this increase in emissions is associated with an increase in wetland emissions due to changes in the East African Lakes (Figure 1).

### 3.2 Annual Increases in South Sudanese CH$_4$ Emissions

Figure 6(a) shows annual CH$_4$ emission estimates from South Sudan between 2010–2016. Our *a posteriori* emissions in the PR1 inversion from the South Sudan region almost doubled from 3.1 (2.4–3.7) Tg yr$^{-1}$ in 2010–2011 to 6.0 (5.2–6.9) Tg yr$^{-1}$ in 2015–2016. Similarly, for the PR2 inversion emissions increased from 3.4 (2.7–4.2) Tg yr$^{-1}$ in 2010–2011 to 6.3 (5.4–7.2) Tg yr$^{-1}$ in 2015–2016. Both inversions indicate a 3 Tg yr$^{-1}$ increase of mean emissions during the inversion period, with

a period of rapid growth between 2011 and 2014. Such a substantial increase in emissions in a short space of time implies a significant change to one or more $CH_4$ sources in this region.

The three main sources of $CH_4$ emissions in South Sudan in our *a priori* inventory are biomass burning, enteric fermentation from agriculture, and wetlands. GFED biomass burning estimates of $CH_4$ for 2010 are 30% smaller than subsequent years,
largely due to a similar decrease in burned area in 2010 (not shown). Although the GFED $CH_4$ estimates therefore increase between 2010 and 2016, this cannot explain the large rise in emissions. *A priori* biomass burning emissions from GFED are estimated to be only 0.25 Tg yr$^{-1}$ or 15% of annual *a priori* South Sudan emissions in 2016, and biomass burning emissions have a distinct seasonal cycle, peaking between November and February, while we observe a $CH_4$ emission increase in all seasons. Figure 6(b) shows that our annual South Sudanese $CH_4$ PR1 emission estimates are driven by changes in all seasons,
but particularly in the wetter months between June–November. A similar pattern was found in the results from the PR2 inversion (not shown). Increased biomass burning may explain some contribution of the increasing emissions during December–February but it cannot explain increases at other times of the year.

Agricultural emissions from enteric fermentation represent another significant source in the *a priori* inventory for South Sudan, accounting for 25% of emissions. South Sudan does host a large livestock population but increased agricultural emissions
are also unlikely to explain the inferred increased in $CH_4$ emissions. Based on IPCC emissions factors for African cattle of 31 kg head$^{-1}$ yr$^{-1}$ (Eggleston et al., 2006), a 3 Tg yr$^{-1}$ increase in emissions would require almost 100 million additional cattle to explain the increase, a population rise equivalent to the entire cattle stocks of the USA. Previous studies have highlighted that the emission factors adopted by the IPCC are highly uncertain and likely too low with values ranging from 31 kg head$^{-1}$ yr$^{-1}$ to 62 kg head$^{-1}$ yr$^{-1}$ for non-dairy cattle (Toit et al., 2014; Wolf et al., 2017; Kouazounde et al., 2014). Even assuming this
upper limit for non-dairy cattle would still require an additional 50 million cattle to explain the rise in South Sudanese emissions. However, livestock estimates from the Food and Agricultural Organization of the United Nations (FAO, available at http://www.fao.org/faostat/) do not indicate a dramatic rise in livestock population, and we do not have any evidence to support the uniform adoption of a higher emission factor for enteric fermentation.

Wetlands are the major source of $CH_4$ in South Sudan, accounting for the remaining 60% of *a priori* emissions, of which
the Sudd wetland, on the course of the White Nile, is the largest, with an area of 9,000–40,000 km$^2$ depending on the season (Rebelo et al., 2012). This wetland area represents a source of 0.8 Tg yr$^{-1}$ in our *a priori* inventory, although our *a posteriori* mean suggests that these emissions are considerably larger (Figure 3), with our mean *a posteriori* estimates being 2.9 and 3.2 times larger than the prior in PR1 and PR2 respectively. To investigate how the Sudd might have changed during the inversion period we use land surface temperature (LST) data from the NASA Moderate Resolution Imaging Spectroradiometer (MODIS)
as a proxy for wetland extent.

The use of satellite LST data as a proxy for soil moisture is well-established (e.g. Cammalleri and Vogt, 2015; Folwell et al., 2016; Gallego-Elvira et al., 2016). In dry conditions, evaporation of water from the soil is restricted, the ratio of latent heat to sensible heat flux decreases and thus the surface temperature increases (Byrne et al., 1979). As such, areas of elevated soil moisture exhibit cooler surface temperatures than dry soil and hence lower LST. The LST anomaly from the climatological
monthly mean LST can thus provide a reasonable proxy for anomalies of the mean seasonal cycle of wetland extent. MODIS

pixels that are wetter than the climatological mean exhibit negative LST anomalies, and the temporal variation of these anomalies indicates the temporal variation in wetland extent. Data were used from both Terra and Aqua MODIS satellites, which have daytime equatorial crossing times of 1030 LT and 1330 LT respectively. To calculate wetland LST anomalies, we used monthly mean daytime-only LST data at 0.05 degree resolution (MOD11C3 and MYD11C3). We identified pixels prone to flooding where the minimum recorded January LST fell below 304 (306) K for Terra (Aqua). To create a monthly anomaly time series, we averaged LST over all of these wetland pixels having first subtracted their monthly climatological values (2003–2018).

Figure 6(c) shows there was a significant decrease in MODIS LST anomalies over areas identified as seasonally flooded in the Sudd between 2010 and 2015, suggesting a growth in mean wetland extent. The decrease in LST anomaly is most pronounced during December—February and March-–May each year, with the seasonal anomalies decreasing by almost 8 K between 2010 and 2015. We find an insignificant trend in LST anomaly during September–November, when wetland extent is greatest. Interpretation of the seasonality of these trends requires some care. During June–November, soils throughout the region are wet, and this strongly suppresses LST, even in the absence of flooding. Moreover, cloud cover is increased during those months, which reduces the reliability of the LST data. The strong signals seen between December and May, when the methodology is expected to be most sensitive to interannual variability in flooding, provides clear evidence of an increase in wetland extent between 2010 and 2015 that helps to explain our increase in *a posteriori* $CH_4$ emissions. The LST data shown before 2010 back to 2007 indicates this was a transient event characterized by a particularly high December–May LST in 2010 (indicating a smaller wetland extent) followed by a trend to a minimum LST in 2015, and not indicative of a longer-term trend.

Our finding is consistent with previous works which have determined an increase in the spatial extent of the Sudd using MODIS LST diurnal temperature difference (Sosnowski et al., 2016), and MODIS land surface reflectance data (Vittorio and Georgakakos, 2018). Both studies indicate that the flooded extent of the Sudd was particularly small in 2009–2010 in both wet and dry seasons, with seasonally flooded vegetation that was constrained to be very close to the White Nile river itself. Vittorio and Georgakakos (2018) reports much greater flooded extents in 2012–2013 and shows an increase in dry season (yearly minimum) wetland extents of around 2000 km$^2$ between 2010 and 2014, related to changes in water flux into the Sudd.

Controls on the wetland extent of the Sudd are dominated by the inflow of water, as evapotranspiration rates exceed direct rainfall for the vast majority of the year (Sutcliffe and Brown, 2018). The inflow to the Sudd is a combination of outflow from the East African lakes (Victoria, Kyoga and Albert, Figure 1), and seasonal variation provided by the runoff from streams in between Lake Albert and the Sudd called the torrents (Sutcliffe and Parks, 1999). The outflow from the East African lakes provides the medium-long term component of the Sudd inflow, while the torrents provide the seasonal peak flooding component. Our finding that *a posteriori* emissions from South Sudan increased at all times of year, together with the trend towards more negative LST anomalies during the dry season are more consistent with an increased upstream inflow from the East African lakes.

The influence of Lake Victoria water levels in particular on the extent of the Sudd has been well documented. Sutcliffe and Parks (1999) estimated that an increase in Lake Victoria water levels in the 1960s led to a trebling of permanent wetland extent in the Sudd, with a smaller effect on the increased extent of seasonal flooding. Recent outflow data from the East African lakes are not publicly available, but they can be approximated using the lake water levels. Since the 1950s the outflow of Lake Victoria

has been controlled by at least one dam, which is regulated by an agreement with countries further downstream on the Nile basin called the Agreed Curve (Sene, 2000). This Agreed Curve is meant to mimic the natural relationship between outflow and lake level, although there is evidence that outflow rates from Lake Victoria far exceeded the Agreed Curve before lake levels reached a minimum in 2006 (e.g. Sutcliffe and Petersen, 2007; Owor et al., 2011; Vanderkelen et al., 2018). However, dam releases after this lake level minimum in 2006 have been shown to be more in line with the Agreed Curve (Owor et al., 2011).

Figure 6 (d) shows satellite altimetry data from Lake Victoria and Lake Albert from the Hydroweb database (Crétaux et al., 2011) between 2007 and 2016. The data show that Lake Victoria annual mean levels rose 0.6 m between 2010 and 2014 and Lake Albert levels rose by 0.8 m in this same period. The increased levels of both lakes imply an increased outflow from the East African lakes which is confirmed by Hydroweb altimetry data of the White Nile at 6.55 °N, 31.40 °E shown in Figure 6 (d). Annual mean water levels at this location at the southern end of the Sudd increased by 0.7 m between 2010 and 2014. Although this latter dataset is affected by variation of inflow to the Sudd due to the torrents, the data show increases of water level in the dry season as well as the wet, indicating the increased water levels at this location at the southern end of the Sudd are likely due to increased flow from further upstream as opposed to seasonal precipitation in the torrents.

In the absence of Lake Victoria outflow data it is possible the rise in lake water levels could be due to human management (through reduced dam releases). However, evaporation not outflow is the major loss process of water from the lake and the positive trends in altimetry data from further downstream at Lake Albert and in South Sudan indicate this explanation is unlikely, since they would not be consistent with a decrease in outflow. Furthermore, the recent increases in Lake Victoria water levels have been attributed to increased precipitation over the Lake Victoria basin (Awange et al., 2019). Whilst there may be no clear positive trend in monthly precipitation data from the Tropical Rainfall Measuring Mission (TRMM, Huffman et al., 2018), Figure 7 shows that increases in Lake Victoria seasonal water height anomalies and associated total terrestrial water storage generally follow large positive seasonal anomalies in precipitation over the lake catchment (delineated by the grey shading). The three years with largest annual precipitation totals over the Lake Victoria catchment between 2001–2016 were 2006, 2007 and 2011, which are all years that correspond to a subsequent increase in both lake levels and GRACE LWE anomalies. The data suggest that the positive precipitation anomalies over the Lake Victoria basin contributed to the rising lake levels and an increase in outflow is therefore likely. Together, the data from these multiple sources indicate increased inflow of the White Nile to the Sudd during the study period, which caused an increase in the extent of wetland flooding and a subsequent substantial increase in $CH_4$ emissions from South Sudan. In common with the LST anomaly data, the altimetry data before 2010 show that the period of rising water levels was largely confined to between 2010–2015, and indicate the transient nature of the increase in inflow and wetland extent, and the subsequent impact on $CH_4$ emissions from South Sudan.

### 3.3 Seasonal Variations of African $CH_4$ Emissions

Figure 8 shows the mean monthly climatological (2010–2016) $CH_4$ emissions from eight selected regions in both NH and SH Africa from the PR1 inversions. With the exception of the Congo basin, the emissions from all other regions shown in Fig. 8 are broadly correlated with the seasonal variation in GRACE LWE anomalies. Chad, Sudan, South Sudan, Madagascar and

the Niger Basin in particular show the strongest correlations with $r^2$ values of 0.7 as shown in Fig. 8. Seasonal variations of *a posteriori* $CH_4$ emissions in most regions are not strongly related to changes in surface skin temperature, which is another parameter commonly used in models to help describe the temporal variation of wetland $CH_4$ emissions (e.g. Gedney, 2004). Seasonal variations in emissions from biomass burning (the timing of which is shown by burned area in Figure 8) are generally

much smaller than the inferred seasonal cycle of emissions. The exceptions are the South Sudan and Angola/Zambia regions. There are two peaks in the seasonal cycle of emissions from South Sudan, the second of which in December coincides with the peak in burned area and associated biomass burning emissions. Similarly in Angola/Zambia there is a noticeable but small emissions peak between July–September when burned area is at a maximum.

Correlations between the seasonality of tropical wetland methane emissions and GRACE LWE anomaly data water storage

from GRACE has been previously reported (e.g. Bloom et al., 2010, 2012). Some of the areas shown in Figure 8 contain significant wetland regions: in South Sudan the Sudd; in Niger Basin the Niger Inland delta; and in Angola/Zambia the Barotse floodplain. Estimates of seasonal wetland extent of the Sudd, suggest that it is composed of around 20% permanent and 80% seasonally flooded wetlands (Rebelo et al., 2012), and as such a large seasonal cycle of $CH_4$ emissions in this region is expected. Similarly, for the Barotse floodplain in Angola/Zambia, past work has estimated an inundated wetland extent that is

approximately ten times larger at its peak compared to the dry season minimum (Zimba et al., 2018). Our 2010–2016 mean *a posteriori* seasonal cycle of $CH_4$ emissions for the Angola/Zambia region (containing the Barotse floodplain) has a peak in January–February that is around 3 times larger than the minimum in October in both PR1 and PR2 inversions, although this region includes emissions from more than this single wetland area.

Taylor et al. (2018) used the Global Inundation Extent from Multi-Satellites (GIEMS, Prigent et al., 2007, 2012) dataset to

identify those wetlands with the greatest seasonal variation in extent, highlighting the Barotse floodplains, the Niger Inland delta and an area to the South of Lake Chad as being most significant. The timing of maximum wetland extent was shown to be largely driven by the seasonal migration of the Intertropical Convergence Zone (ITCZ), with peak extents in NH Africa typically around October. Figure 8 shows the seasonal peak in our *a posteriori* northern hemisphere emission regions (panels (a)–(e)) occurs consistently between September and November. If we assume that this is largely driven by the seasonal cycle of

wetlands then the timing of our *a posteriori* emissions peak is more consistent with the findings of Bloom et al. (2012), inferred from SCIAMACHY $XCH_4$ data, where northern tropical Africa emissions peak between September–November, compared to the later work of Bloom et al. (2017) in which the seasonal cycle peaks earlier in April–October.

We acknowledge that the large inferred seasonal cycles shown in Figure 8 may not reflect exclusively changes in wetland emissions. Based on production estimates, rice paddy emissions will follow a similar seasonal cycle (Laborte et al., 2017),

although emissions from rice comprise only 4% of the *a priori* anthropogenic component of tropical African emissions, and are largely concentrated in West Africa. In addition, the contribution of livestock enteric fermentation emissions to the seasonal $CH_4$ emission cycle cannot be discounted. In the semi-arid ecosystems of sub-Saharan Africa, ecosystem productivity is strongly linked to soil moisture (Madani et al., 2017). Studies of sub-Saharan cattle have shown that animal weights and dry matter intake vary seasonally, linked to the availability of forage (Ayantunde et al., 2005; Assouma et al., 2018). As such, the

seasonal variation of enteric fermentation $CH_4$ emissions from livestock could be a significant contributor to the seasonal cycle

of total CH$_4$ emissions that we infer. One potential indicator of this is that we infer large seasonal cycles in regions that contain the largest livestock populations, such as Ethiopia and Sudan. The magnitude of the seasonal cycle in Ethiopia is comparable to that of South Sudan, despite the former not containing seasonal wetlands of the extent of the Sudd. However, since we are limited to resolving emissions at broad scales, and both agricultural and wetland sources have similar spatial distributions across Africa, we are unable to quantify the relative contributions of different sources to the seasonal cycle.

Our results for the Congo Basin show no evidence of a significant water storage dependent seasonal cycle of CH$_4$ emissions. The Congo basin straddles the equator and therefore experiences two wet seasons each year, in March and October (Figure 8). The central part of the basin, referred to as the Cuvette Centrale, is permanently flooded, largely rain-fed and contains large stores of peat (Dargie et al., 2017). These features are consistent with a muted seasonal cycle of CH$_4$ emissions. In contrast with the *a priori* emissions we use from WetCHARTs ensemble mean, which have an average peak-to-peak seasonal cycle magnitude of around 4 Tg yr$^{-1}$, our *a posteriori* emissions mean peak-to-peak seasonal cycle is $\simeq$1 Tg yr$^{-1}$. This seasonal cycle magnitude is smaller than the mean monthly 95% uncertainty range on the *a posteriori* estimates. Our results from the Congo Basin are consistent with the results reported by Parker et al. (2018), who identified a smaller peak-to-peak seasonal cycle in GOSAT XCH$_4$ data compared to the XCH$_4$ predicted by the WetCHARTs ensemble mean.

## 3.4 Sensitivity to model bias correction

In addition to the sensitivity to different proxy XCH$_4$ datasets, we also tested the sensitivity of our results to a bias in the GEOS-Chem model that has previously been reported for CH$_4$ (Turner et al., 2015; Maasakkers et al., 2019). We applied the quadratic regression bias correction $y = 0.005 \times (\lambda^2 - \lambda - 100))$ from Turner et al. (2015) to the PR1 data and performed a new set of inversions, (referred to as PR1$_{BC}$) where $\lambda$ is the latitude in degrees, and $y$ the bias term. The bias correction was around 3 ppb at the north and south boundaries of our tropical domain, tending to –0.5 ppb at the equator.

The results of this bias corrected inversion show the mean total African emissions to be 72 (70–74) Tg yr$^{-1}$, compared to 76 (74–78) Tg yr$^{-1}$ in the non-bias corrected PR1 inversion. The difference in total African emissions is equivalent to the difference between PR1 and PR2 inversions, albeit with smaller total emissions when the bias correction was applied. The trend in tropical African emissions in the PR1$_{BC}$ inversion was 2.0 (1.6–2.4) Tg yr$^{-1}$, compared to 2.1 (1.7–2.5) Tg yr$^{-1}$ in the PR1 inversion. In common with the PR1 inversion the trend in East African emissions was also 1.7 (1.4–2.0) Tg yr$^{-1}$ in PR1$_{BC}$. Emissions from the Sudd were found to have increased at a rate of 0.3 Tg yr$^{-1}$ in PR1$_{BC}$. As such, our main conclusions in this work are robust to the presence of a small latitudinally dependent bias in the GEOS-Chem simulation of CH$_4$. We propose this is because the presence of any latitudinal bias can be largely subsumed into the boundary conditions scaling in the regional inversion, minimizing the impacts of a potential model transport bias.

## 4 Concluding Remarks

We use GOSAT proxy XCH$_4$ data and a nested version of the GEOS-Chem atmospheric chemistry transport model to infer emissions of CH$_4$ over tropical Africa. At the heart of this data product is the ratio of XCH$_4$:XCO$_2$, which effectively minimizes

spectral artifacts due to cloud and aerosol scattering. A model estimate of $XCO_2$ is typically used to infer a proxy retrieval of $XCH_4$. In this work, we use two $XCO_2$ model estimates: 1) an ensemble mean of three independent models (including GEOS-Chem) denoted as PR1; and 2) GEOS-Chem fields that have been fitted to GOSAT $XCO_2$ denoted as PR2. Mean *a posteriori* tropical African emission estimates for 2010–2016 are 76 (74–78) Tg yr$^{-1}$ and 80 (78–82) Tg yr$^{-1}$ for PR1 and

PR2, respectively. Our results illustrate the sensitivity of *a posteriori* $CH_4$ emissions on our choice of $XCO_2$ to determine the proxy $XCH_4$ data product. Here, the difference is driven by a seasonal cycle of $CO_2$ fluxes over northern tropical Africa inferred from GOSAT that is larger than that inferred by *in situ* data.

Our *a posteriori* $CH_4$ emissions represent 15% of the global emissions estimate of 546 Tg yr$^{-1}$ in Maasakkers et al. (2019) derived from GOSAT data between 2010 and 2015. Whilst they represent a significant fraction of the global budget, changes

over tropical Africa are unlikely to be exclusively responsible for observed global-scale variations. We find a mean emissions trend of 2.1 (1.7–2.5) Tg yr$^{-1}$ from PR1 and 1.5 (1.1–1.9) Tg yr$^{-1}$ from PR2 between 2010 and 2016, representing around a third of the global growth trend in emissions during this period from global inversions. McNorton et al. (2018) reported a global growth rate between 2007–2015 of 5.7±0.8 Tg yr$^{-1}$. Thompson et al. (2018) found an increase in global microbial sources of 24–48 Tg yr$^{-1}$ between 2006–2014 mostly originating from tropical latitudes, equivalent to a global growth rate

of 3–6 Tg yr$^{-1}$. Although Saunois et al. (2017) found emissions from the tropics to have increased by 18 (13–24) Tg yr$^{-1}$ between the periods 2002–2006 and 2008–2012, relatively little of this was ascribed to Africa. This is qualitatively consistent with our hypothesis that the increase in emissions from East African regions, and in particular the Sudd, due to increased water levels of the East African lakes was limited to the period between 2010 and 2016. Moreover, previous studies have tended to focus on large continental regions when investigating trends in $CH_4$, which may mask any smaller regional changes that are

only apparent through inspecting finer-scale spatial distributions. For comparison to other global regions, our *a posteriori* trend for tropical Africa is larger than those recently reported for China of 1.1 (0.7–1.5) Tg yr$^{-1}$ and India (0.7 (0.2–1.2) Tg yr$^{-1}$) between 2010 and 2015 (Miller et al., 2019).

We attribute a large part of the increase in African emissions between 2010 and 2016 to the increasing wetland extent of the Sudd, driven largely by increased water levels in the upstream East African lakes. Emissions from the Sudd wetlands are

found to have increased during the study period by 3 Tg yr$^{-1}$. However, satellite altimetry data shows a stabilisation of the East African lake levels after 2014, and LST anomaly data, used as a proxy for wetland extent, suggest the trend of increasing Sudd wetland extent only lasted between 2010–2015, and that the wetland extent was anomalously small in 2010. Our *a posteriori* emission estimates also suggest a stabilisation of emissions in 2015–2016, indicating that the increase in emissions from the Sudd was a transient event. No other easily identifiable wetland areas were estimated to have a significant growth in emissions

between 2010 and 2016. Continental-scale atmospheric isotopic $^{13}CH_4$ data show a shift towards lighter values, which have been used to suggest an increase in tropical microbial emissions since 2007 (Nisbet et al., 2016; Schaefer et al., 2016). Our findings over South Sudan are not inconsistent with this hypothesis. Although our inversions do not distinguish between different emission source of $CH_4$, the location and trends in satellite LST and altimetry data suggest this increase was due to wetland emissions from the Sudd. FAO estimates of livestock population and IPCC Tier 1 emission factors also suggest a

modest increase in agricultural emissions from the wider East Africa region of 0.2 Tg yr$^{-1}$, which would also be qualitatively consistent with the trends in isotope signals.

Our *a posteriori* emission estimates show that $CH_4$ emissions across tropical Africa are highly seasonal, with the peak in monthly emissions of each hemisphere strongly correlated with the peak in ground water storage. Whilst the link between water storage and wetland $CH_4$ emissions is well established, the lack of a seasonal cycle of emissions in the Congo Basin highlights that this linkage is not uniform across all tropical wetlands. Furthermore, the presence of a large seasonal cycle in regions such as Ethiopia where livestock are likely to be the dominant source may indicate seasonality in other $CH_4$ sources. However, emissions were resolved using coarse basis functions largely at national scales and at monthly timescales which do not allow us to separate out the contribution of different sources. Future studies using higher resolution data from satellites such as TROPOMI may help to better understand the temporal and spatial characteristics of the seasonal cycle of methane emissions from both wetlands and other sources. Indeed, initial data from TROPOMI appears promising in isolating large wetland systems such as the Sudd (Hu et al., 2018), which should enhance the future monitoring of this large $CH_4$ source region. Incorporating additional information from complementary measurement data into the inversion system could further help to reduce uncertainties in the $CH_4$ budget. Examples of potentially useful space-based measurements include formaldehyde to constrain the temporal and spatial variability of the OH radical sink (Wolfe et al., 2019), and GRACE data (and its follow-on mission) to help constrain the temporal variability of wetland emissions.

Satellite observations now play an integral role in observing and monitoring land, ocean, and atmospheric components of the global carbon cycle (Palmer, 2018). As these data become available at progressively higher spatial resolution, we can begin to address scientific questions that focus on understanding how different ecosystems change with climate. However, the value of these data disproportionately increases with the availability and integration of *in situ* atmospheric and ecological data that provide complementary detailed information on finer spatial and temporal scales.

*Data availability.*  UoL GOSAT satellite column observations of $CH_4$ are available for download from the Centre for Environmental Data Analysis, http://catalogue.ceda.ac.uk/uuid/f9154243fd8744bdaf2a59c39033e659.

*Code and data availability.*  The GEOS-Chem model is a community model and is freely available (http://acmg.seas.harvard.edu/geos/). The model metadata is freely available.

*Author contributions.*  MFL and PIP designed the experiments, and MFL performed all calculations. MFL wrote the paper with contributions from PIP. LF provided global $CH_4$ and $CO_2$ inversion outputs. CMT provided inputs on the hydrological cycle and LST over tropical Africa. HB and RJP provided access to the GOSAT XCH$_4$ data. All coauthors provided comments on the manuscript.

*Competing interests.* The authors declare no conflicts of interest.

*Acknowledgements.* MFL and PIP gratefully acknowledge funding from the Methane Observations and Yearly Assessments (MOYA) project funded by the National Environment Research Council (NERC, grant #NE/N015916/1). PIP, HB, RJP and CMT acknowledge support from the UK National Centre for Earth Observation (NCEO). NERC provides national capability funding to NCEO (grant #PR140015). HB and RJP acknowledge funding from Copernicus C3S and ESA CCI.

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

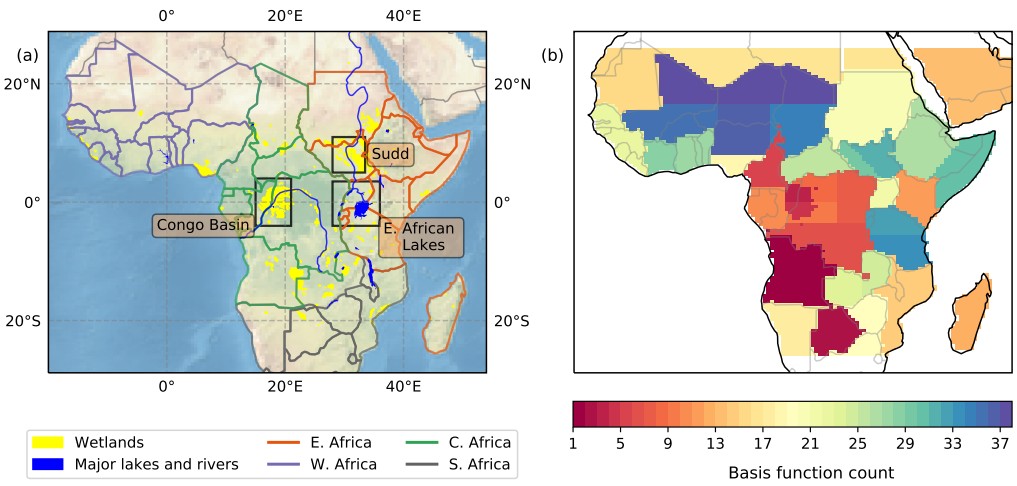

**Figure 1.** (a) Map of our tropical African model domain. Wetland regions from Gumbricht et al. (2017) are marked in yellow and major rivers and lakes are marked in blue. The Congo Basin, the Sudd, and the East African Lakes are outlined by black boxes. The division into regions of the African Union are shown by the different colour country borders, indicating Central Africa (green), East Africa (orange), Southern Africa (grey) and West Africa (purple). The map background represents shaded relief from https://www.naturalearthdata.com; (b) The spatial emission basis functions used in our inversions.

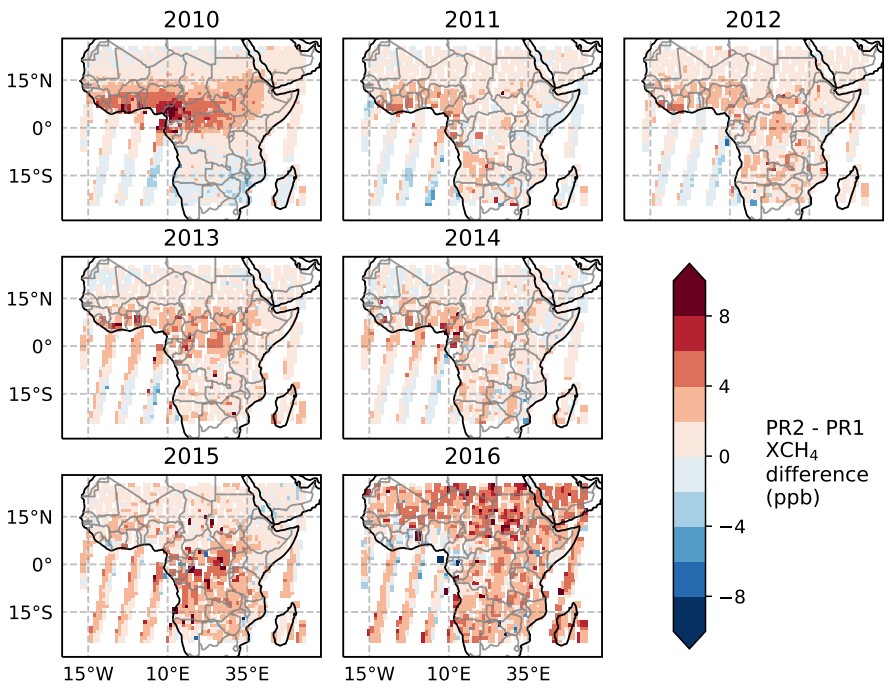

**Figure 2.** Annual mean differences between PR1 and PR2 proxy $XCH_4$ data at $1°x1°$ resolution. PR1 represents the data from Parker et al. (2015). PR2 uses our own $CO_{2\ model}$ component, generated from an inversion of GOSAT $XCO_2$ data to form the proxy $XCH_4$ data.

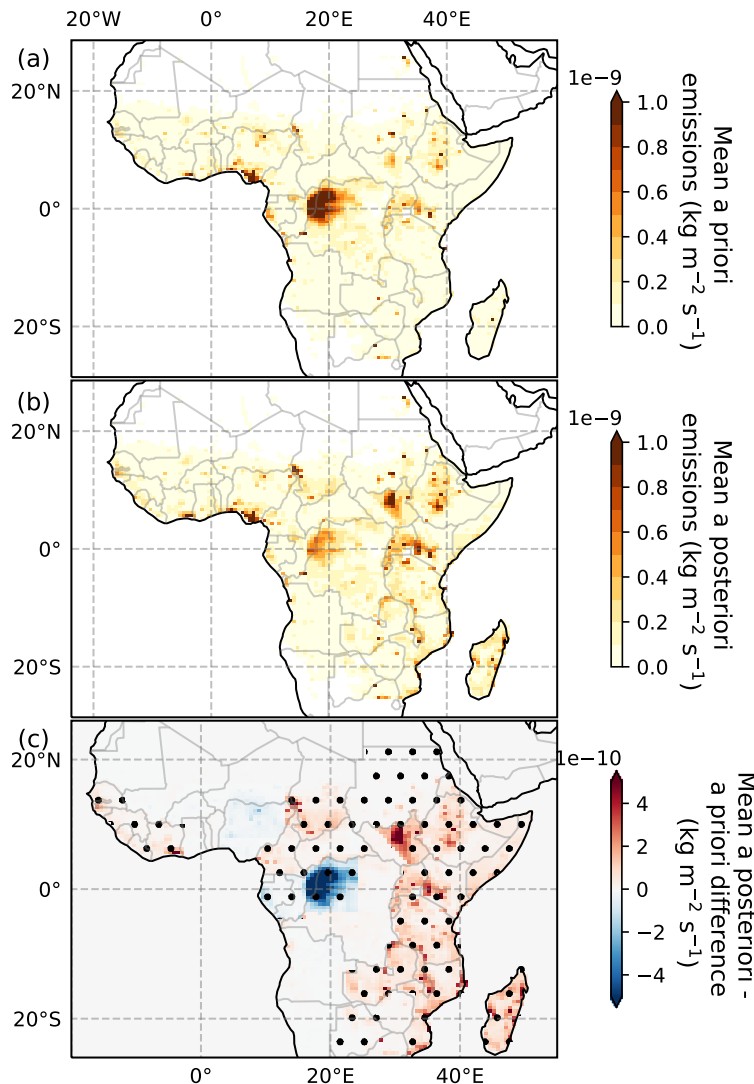

**Figure 3.** The 2010–2016 mean (a) *a priori* and (b) *a posteriori* spatial distribution of CH$_4$ emissions in the tropical African domain from the PR1 inversion. (c) Mean PR1 *a posteriori* minus *a priori* differences, 2010–2016. Stippling indicates areas where the *a priori* mean was outside the *a posteriori* 95% uncertainty range for both PR1 and PR2 inversions.

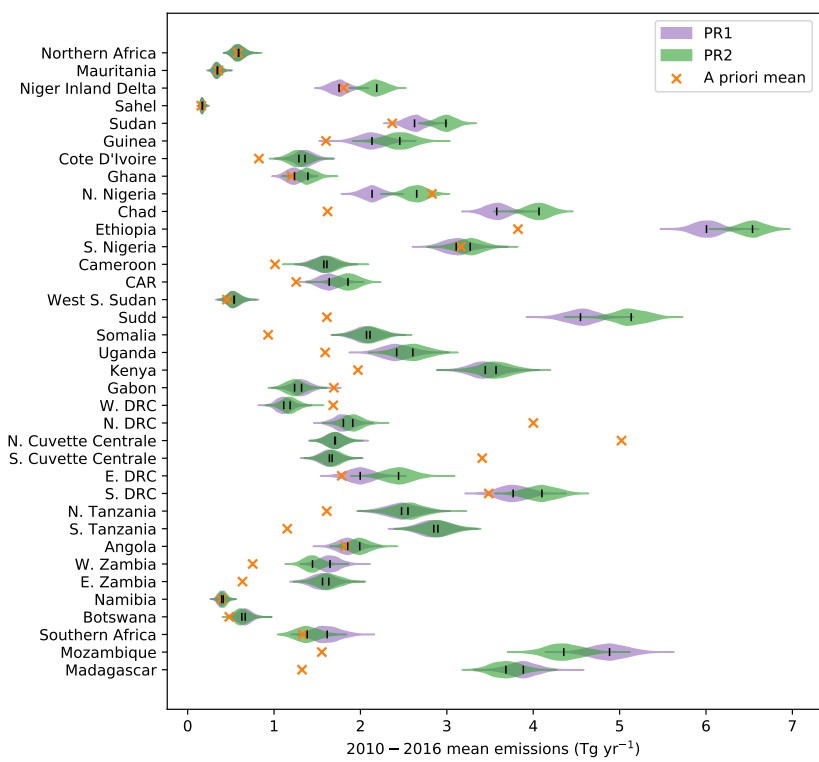

**Figure 4.** Violin plot of the *a posteriori* distribution from each African basis function for the PR1 (purple) and PR2 (green) inversions over 2010–2016. The lines in the centre of the violins represent the mean of each respective distribution. The *a priori* mean value for each basis function region is shown by the orange cross. Basis function regions are ordered from South to North travelling up the y-axis. The region names on the y-axis are intended to be representative of the respective regions and not necessarily an exact definition based on country borders. CAR is the Central African Republic. DRC is the Democratic Republic of Congo.

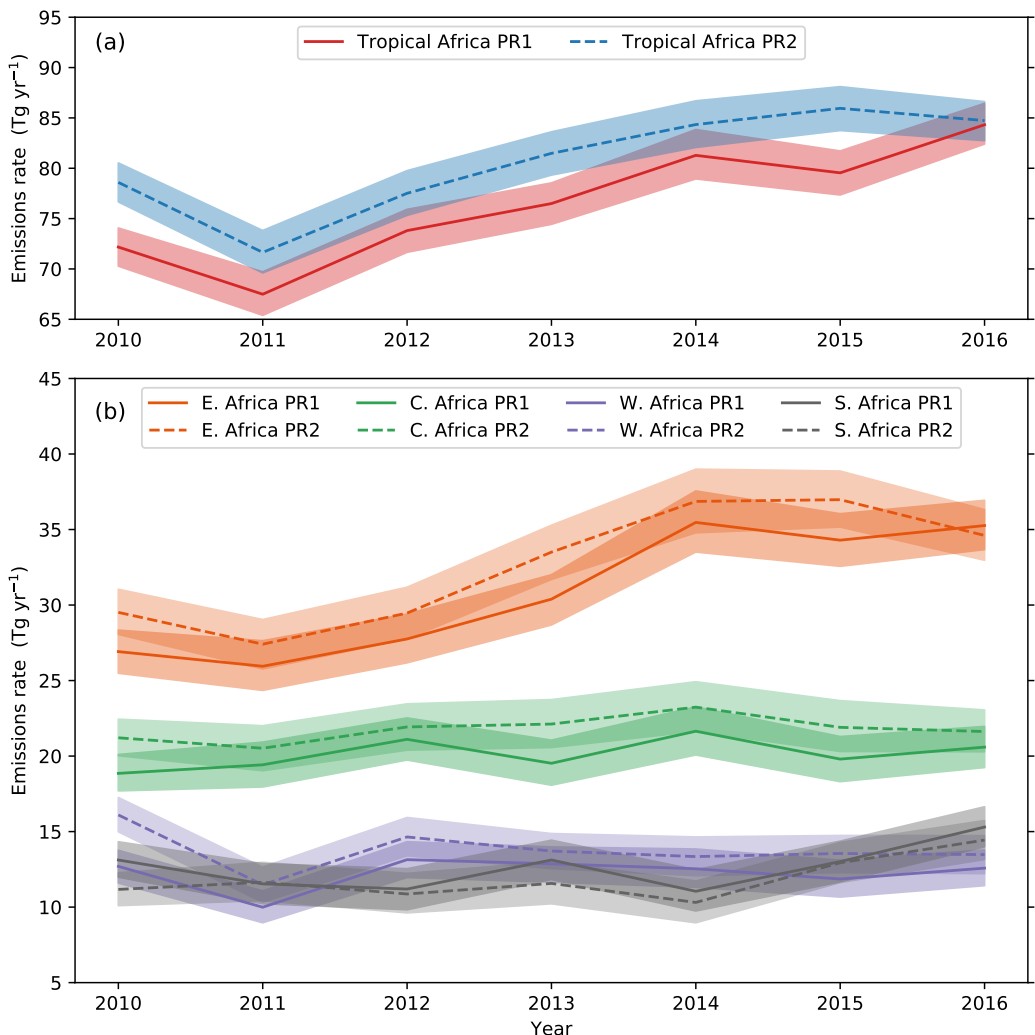

**Figure 5.** (a) Annual *a posteriori* emissions for tropical Africa. (b) Annual *a posteriori* emissions for East Africa (blue), Central Africa (purple), West Africa (orange) and Southern Africa (grey). Solid lines represent the PR1 *a posteriori* mean, dashed lines indicate the PR2 *a posteriori* mean in each panel. Shading represents the 95% uncertainty range in both panels.

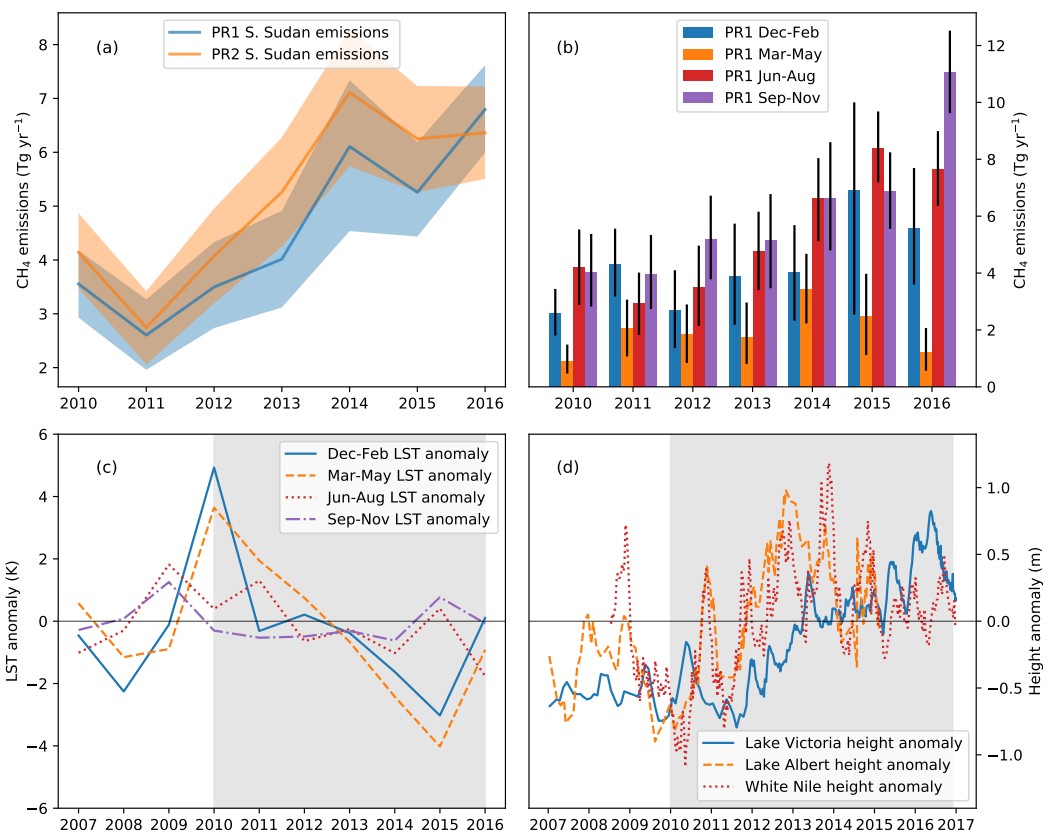

**Figure 6.** (a) Annual mean *a posteriori* emissions from the area encompassing the Sudd in South Sudan from PR1 (blue) and PR2 (orange) inversions. Shading represents the 95% uncertainty range. (b) The annual mean of Sudd emissions in each season from the PR1 inversions. Uncertainty bars represent the 95% uncertainty range. (c) The annual mean LST anomaly in each season from the Sudd, used as a proxy for wetland extent between 2007–2016. Negative anomalies indicate increased soil moisture. The grey shading indicates the inversion period when the decrease in LST anomalies occurs. (d) Satellite altimetry data from Lake Victoria (blue), Lake Albert (orange) and the White Nile at 6.55 °N, 31.40 °E (red) between 2007–2016, showing the transient increase in upstream water levels during the inversion period (grey shading).

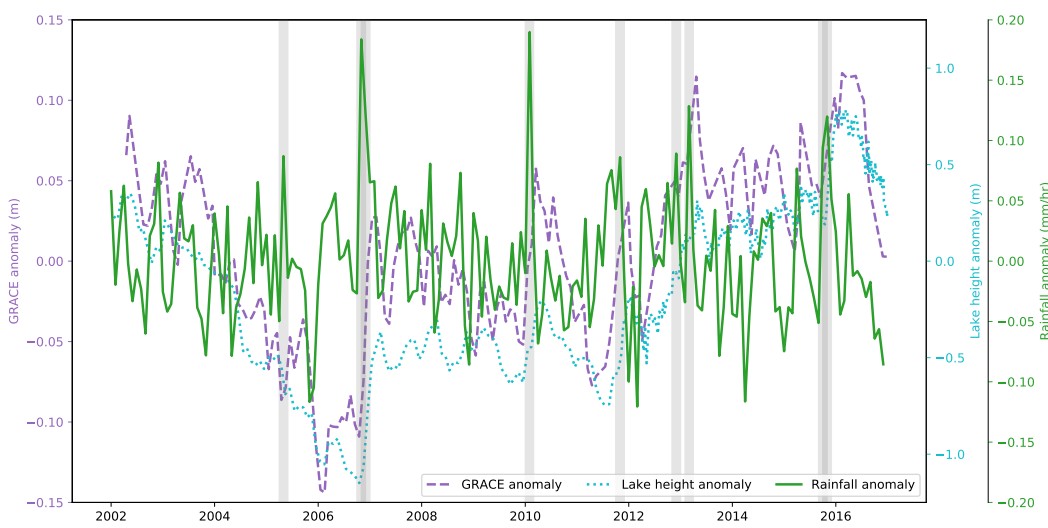

**Figure 7.** Monthly anomalies of precipitation over the Lake Victoria basin from the Tropical Rainfall Measuring Mission (TRMM, green), GRACE LWE over the basin (purple) and Lake Victoria water height (cyan). Anomalies represent the departures from the mean of each month between 2002–2016, to remove seasonal cycle influence. Grey shading indicates 3-month periods centred on months where the precipitation anomaly was greater than the 95th percentile. These periods coincide with rises in both LWE and lake height anomalies, highlighting the role of heavy rainfall on lake level increases.

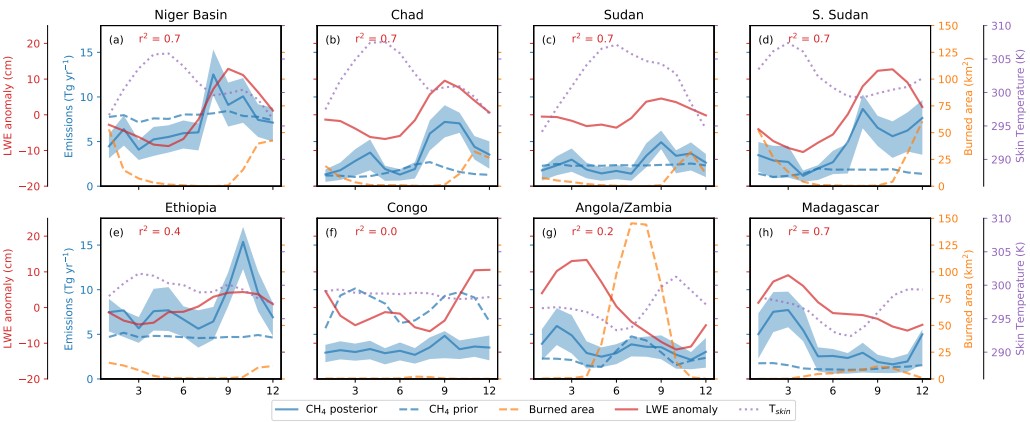

**Figure 8.** Mean seasonal cycle of *a posteriori* CH$_4$ (solid blue) emissions from eight regions across tropical Africa. Shading represents the 95% uncertainty range. Also shown are the *a priori* emissions (dashed blue), burned area (orange dashes), GRACE liquid water equivalent height anomaly (LWE, red), and surface skin temperature (T$_{skin}$) estimates from MERRA-2 (purple dots). Plots (a)–(e) represent regions in northern hemisphere Africa with CH$_4$ emissions peak in the latter part of the year and (g)–(h) are in southern hemisphere Africa where the CH$_4$ emission peak is at the start of the year. (f) Congo straddles the equator. The r$^2$ value represents the correlation coefficient between *a posteriori* emissions and LWE values.