# Peer review of "An increase in methane emissions from tropical Africa between 2010 and 2016 inferred from satellite data"

_Atmospheric Chemistry and Physics, 2019_

## Referee Comment (RC1) · Anonymous Referee #2 · 9 Jul 2019

**General comments**

Lunt et al. constrain methane emissions in tropical Africa with GOSAT $XCH_4$ data to retrieve monthly $CH_4$ emissions in the framework of a Bayesian inversion. A very useful sensitivity test (to the $CO_2$ fields used in the $XCH_4$) is included, which makes it possible to provide uncertainty ranges for the estimates of emissions and comment on the significance of the results. With satellite $XCH_4$ data from 2010 to 2016, they are able to study seasonal cycles as well as the 7-year trend. Using various other satellite data (land surface temperature anomaly, altimetry), they are able to suggest links between their findings regarding $CH_4$ emissions and the sources contributing to the variations

of these emissions, which are mainly wetlands in this area. This study provides esti-mates for $CH_4$ emissions and their variations at various spatial and temporal scales in an area, tropical Africa, which is both a key-region for methane emissions and a region where these emissions are very uncertain. The authors have been able to exploit not only satellite data of $XCH_4$ to assimilate in their inversion framework, which already provides rich insights regarding methane emissions, but also to proceed further in the investigation of the possible drivers of the variations of these emissions, making use of different satellite data sets. The manuscript is clearly written and well-structured, the results are very clearly presented and discussed. For all these reasons, I recom-mend publication of this manuscript in ACP after minor corrections (most corrections for Section 2.3).

**Specific comments**

p.2 l.31: what about wild animals for enteric fermentation?

p. 2 l.33: indicate what the range corresponds to (one-sigma, 95%, the full range of the ensemble) so that the reader can compare to the other ranges in the paper.

p.4 l.4: explanation for the year 2016?

p.4 l.20: "as opposed to XCO2": I don't understand the idea implied here.

p.6, Section 2.3: I think this Section is a bit confusing as it seems to be trying to explain a methodology but not in details so that some useful elements are missing for the reader and others are presented in too general a way. Some particular remarks below. Could you try re-writing the Section with a kind of clear hierarchy in the important points for understanding the study described in the paper and the general points which are detailed in the references?

p.6 l.28: "reducing the impact of underlying assumptions": I think it is a bit optimistic to put it like this. The assumptions are still there but they are not made on the same variables/parameters. For instance, the assumption that the errors can be specified as

PDFs and, then, the user's choice of one given type and form (be it Gaussian, Poisson or another).

p.6 l.30: "uncertainties in the a posteriori distribution are more representative of uncertainties in the inversion system": more representative than what?

p.6 l.31: "any form of error variances can be used": do you mean covariances? "Any form" is misleading: they still need to be PDFs, probably ones defined by only a few parameters.

p.7 l.12: "all parameters in turn": does it mean one parameter per iteration?

p.7 l.13: which are the hyper parameters here?

p.7 l.27: what is the type and form of the posterior distribution?

p.7 l.20: reference for the measurement uncertainty? Same in both XCH4?

p.7 l.23: can you comment a bit on the validity of the "uncorrelated" assumption?

p.8 l.11: you should also report the range for the Saunois et al. figure, so that the following conclusion (l. 15-16) that your results are consistent with it is actually supported. You can also make use of Saunois et al. 2017: https://doi.org/10.5194/acp-17-11135-2017

p.8, l.17: Fig 3: it is difficult to interpret the fact that the prior is outside the posterior 95% uncertainty range without any information on the types and forms of the PDFs. Maybe something like Fig 4 of http://dx.doi.org/10.1016/j.spasta.2016.06.005 could be useful?

p.9, Section 3.1: make use of Saunois et al. 2017: they discuss the variability over 2000-2012 so that there are only three years common with your period but you could put your trends in perspective.

p.9 l.13: "although there was substantial inter-annual variability": I don't understand the

logical link between the inter-annual variability (which can go either way from one year to the other) and the trend.

p.9 l.22: why are the glint data more sensitive to the boundary conditions?

p.11 l. 4-6: indicate whether the link between the variations of LST anomalies and wetland extent variations is a (reasonable) assumption or a proven proxy link. Maybe with a reference?

p.13 l.4: "$r^2$ values of 0.2-0.8": 0.2 does not seem to be such a strong correlation. Do you have criteria for the significance of this?

p.15. l.1 seq: make use of Saunois et al. 2017.

**Technical corrections**

fig 1 (a): what is the background of the map: climatological vegetation cover?

p.3 l.19: "Gravity Recovery and Climate Experiment (GRACE," -> And

p.3 l.20: "liquid water equivalent height (LWE) height anomaly retrievals" -> delete first height

p.4 l.31: "different to" -> different from?

p.7 l.28: "clearly show" -> clearly shows

p.8 l.17 seq.: there are a lot of numbers in the two paragraphs: would it be possible to make a table? E.g. with columns prior, PR1 posterior, PR2 posterior. Same remark for p.9 l.2 seq., p. 10 first paragraph

p.8 l.33: "that" -> than (smaller l. 32)

p.9 l.12: "reigon" -> region

p.10 l.18: "represents" -> represent

p.11 l.4: "anomalies from" -> "anomalies of"?

p.11 l.30: missing )

p. 15 l.32: "changes" -> change

---

## Referee Comment (RC2) · Anonymous Referee #1 · 24 Jul 2019

**1  Overview:**

Review of "*An increase in methane emissions from tropical Africa between 2010 and 2016 inferred from satellite data*" by Lunt *et al.*

Lunt *et al.* present an analysis using 7 years of GOSAT methane measurements over Africa. They use these measurements in a hierarchical Bayesian inference framework to estimate monthly methane emissions from Africa. The authors then employ a number of correlative measurements (e.g., land surface temperature and water levels in Lake Victoria) to deduce the underlying drivers of these methane emissions. Overall, I

think the work is excellent. The text is clear and reasonably concise, figures are generally high quality, and my comments are all seemingly minor. I suggest the paper be accepted pending minor revisions.

**2 Minor Comments:**

2.1 Bias correction term for the model or GOSAT?

The only true shortcoming I found in the paper was the lack of a bias correction term. Most papers using GOSAT data have investigated the possibility of a latitudinally dependent bias. Fraser et al. (2014), Alexe et al. (2015), and Turner et al. (2015) all included some sort of polynomial or quadratic bias correction term (although some attributed it to the model stratosphere). Others used a bias term that was dependent on air mass factor (Cressot et al., 2014), that would also lead to latitudinal differences. The authors study domain covers $\sim 50°$ of latitude, so it seems like this could be an important factor. Including this in the hierarchical inversion framework would be quite exciting.

2.2 $CO_2$ fields for the proxy method

It would be illustrative to also show a comparison between PR1, PR2, and a third case where the proxy retrievals are constructed using the full-physics $XCO_2$ retrieval:

$$XCH_{4\,proxy} = \frac{XCH_{4\,no-scatter}}{XCO_{2\,no-scatter}} \times XCO_{2\,full-physics} \qquad (1)$$

This could be included in Figure 2, I don't think it's necessary for the authors to perform an additional inversion with this retrieval. It would, however, be nice to show a retrieval

that is independent of modeled $CO_2$ fields.

**2.3 Number of MCMC samples**

Page 7, Line 10 mentions that the *a posteriori* distributions each have 2,000 samples. However, aren't there 3,697 basis functions? Does that mean there are less than one sample per basis function? Do these posterior distributions really sample the full space?

**2.4 Mention 2010 LST anomaly discussion in the abstract**

I think it's important that the authors mention that 2010 may be an anomalously low year for wetland emissions in the Sudd (based on their LST analysis). This paper will likely get quite a bit of attention because a number of groups are looking for trends in wetland emissions. The authors do an excellent job of discussing the nuances of their trends in the main text and conclusions, but I think there should be a short (less than one sentence) mention of this possible anomaly in 2010.

**2.5 Wolfe et al.**

The authors should mention Wolfe et al., *PNAS* (2018): "Mapping hydroxyl variability throughout the global remote troposphere via synthesis of airborne and satellite formaldehyde observations" and consider using this data to help constrain their OH fields in the future (would be beyond the scope here).

[Figure]

**3 Specific comments:**

Page 2, Line 23: Suggest replacing *mathematically* with *mechanistically*.

Page 3, Line 11: Suggest replacing *strong signatures* with a synonymous phrase. I typically associate "signatures" with isotopic source signatures, which are also discussed in the manuscript.

Page 5, Lines 8–12: This paragraph seems out of place, I feel like it should go before the discussion of $CO_2$ fields. Right now it goes $->$ GOSAT details $->$ proxy retrievals $->$ $CO_2$ fields $->$ impact of $CO_2$ fields $->$ back to GOSAT details.

Page 5, Line 31: I think you should move the Feng et al. (2017) citation to the beginning of Line 30. It initially sounded like *you* were building some Kriged global concentration fields from the NOAA data because of this text "...coarser global run that was **fitted** to *in situ* data..."

Page 6, Line 4: Suggest replacing "prior" with "*a priori*" for consistency with the rest of the text.

---

## Author Comment (AC1) · 1 Oct 2019

We thank both reviewers for their comments. Below we respond to all individual reviewer comments. Reviewer comments are denoted by italicized text.

**Reviewer 1**
**1 Overview:**

> *Lunt et al. present an analysis using 7 years of GOSAT methane measurements over Africa. They use these measurements in a hierarchical Bayesian inference framework to estimate monthly methane emissions from Africa. The authors then employ a number of correlative measurements (e.g., land surface temperature and water levels in Lake Victoria) to deduce the underlying drivers of these methane emissions. Overall, I think the work is excellent. The text is clear and reasonably concise, figures are generally high quality, and my comments are all seemingly minor. I suggest the paper be accepted pending minor revisions.*

We thank the Reviewer for their assessment of the work and address their specific comments below.

**2 Minor Comments:**

> *2.1 Bias correction term for the model or GOSAT?*

> *The only true shortcoming I found in the paper was the lack of a bias correction term. Most papers using GOSAT data have investigated the possibility of a latitudinally dependent bias. Fraser et al. (2014), Alexe et al. (2015), and Turner et al. (2015) all included some sort of polynomial or quadratic bias correction term (although some attributed it to the model stratosphere). Others used a bias term that was dependent on air mass factor (Cressot et al., 2014), that would also lead to latitudinal differences. The authors study domain covers ~50° of latitude, so it seems like this could be an important factor. Including this in the hierarchical inversion framework would be quite exciting.*

This is an issue we did indeed consider when setting up the inversion. However, we do not think that the inclusion of a bias term in the GOSAT data is necessary in this work. The Turner et al (2015) paper highlights this latitudinal bias but shows that the bias is most pronounced at high latitudes and "becomes significant at latitudes polewards of 50". A similar point is made in Maasakkers et al (2019). Since our study area covers between 26S – 26N the bias is anticipated to be small (less than 5 ppb) and impacts of such a bias on derived emissions may not be as great as at higher latitudes.

Furthermore, a bias in background concentrations may be absorbed into the boundary condition terms in the regional inversion, since scale factors applied to these background fields were allowed to vary and solved for in the inversion. This is unlike a global inversion where we agree

any latitudinally dependent errors in model transport would make this bias correction term important.

Nevertheless, to address this reviewer's comment we have tested our prior assumptions about the importance of this bias at low latitudes by reporting further calculations. We have done so by applying the quadratic regression bias correction, $y = 0.005 * (x^2 - x - 100)$, from Turner et al. (2015) to the data (where y is the bias and x the latitude in degrees) and rerunning the PR1 inversion. This implies a bias of around 3 ppb at the N and S boundaries of our domain, tending to $-0.5$ ppb at the equator.

Total African emissions using PR1 data with the Turner bias correction applied are 72 (70-74) Tg yr$^{-1}$ with a trend of 2.0 (1.6-2.4) Tg yr$^{-1}$. For comparison, without the bias correction emissions are 76 (74-78) Tg yr$^{-1}$ for the PR1 inversion with a trend of 2.1(1.7-2.5) Tg yr$^{-1}$. The East Africa trend with the bias correction is 1.7 (1.4-2.0) Tg yr$^{-1}$, and without is 1.7 (1.4-2.0) Tg yr$^{-1}$. The Sudd emissions trend with the bias correction is 0.3 Tg yr$^{-1}$, with emissions increasing from 2.9 Tg yr$^{-1}$ in 2010-2011 to 5.7 Tg yr$^{-1}$ in 2015-2016, and without the bias correction emissions are 0.4 Tg yr$^{-1}$, with emissions increasing from 3.1 Tg yr$^{-1}$ in 2010-2011 to 6.0 Tg yr$^{-1}$ in 2015-2016.

We also reran the PR1 inversion using the bias correction term from Maasakkers et al (2019) which is very similar to that of Turner et al (2015) of the form: $y = 0.001 * (4x^2 - 1.3x) - 5$. The results from this test were almost identical to results from applying the Turner et al. (2015) bias correction, despite a different minimum (-5 vs $-0.5$). However, this absolute offset to all mole fractions can be absorbed in the boundary condition scaling, which we include as part of our state vector, and the shape of the latitudinal dependence between the two bias corrections is otherwise very similar.

The inclusion of a bias term has an impact on the total African emissions, reducing them by 4 Tg yr$^{-1}$, likely due to boundary condition scale factors needing to be slightly larger to account for the higher observed background mole fractions. However, the difference in the derived trends for Africa, East Africa and the Sudd are small and the main conclusions we draw in this work are unaffected. We propose this is for a number of reasons:

A) The region of interest is concentrated in the tropics where the change in bias term applied in Turner et al (2015) and Maasakkers et al (2019) was relatively small (less than 3 ppb).

B) We have 6 scaling terms for the boundary conditions. (N, NE, SE, S, SW and NW). As such, a latitudinal bias in air incoming to the domain could be accounted for by adjustments to these boundary conditions in the inversion.

C) The bias term is thought to be a stratospheric transport bias in the GEOS-Chem model, which is particularly evident at 4x5 degree resolution. Stanevich (2018) showed that the difference between GOSAT and GEOS-Chem is greatly reduced when a global simulation is performed at 2x2.5 degrees, and Maasakkers et al (2019) showed this difference is mostly in the simulation of

the stratosphere. Assuming that the bias magnitude is dependent on the model resolution, it could be further reduced at 0.5 x 0.625 used in the regional modelling of this work.

Since the bias may be resolution dependent, the 4x5 bias diagnosed by Turner et al (2015) and Maasakkers et al (2019) may not be appropriate for the higher resolution used here. Furthermore, since the identification of such a bias at 0.5x0.625 would be heavily dependent on the accuracy of emissions within the domain, we do not think it appropriate to apply an ad-hoc bias correction term to the GOSAT data in this work.

Solving for a latitudinally dependent bias term might be possible within the hierarchical approach, although it is not clear to us whether this may just act to counteract changes to the boundary condition scaling, such that neither parameter would obtain convergence within the MCMC framework.

In response to this comment we have included the results of the bias correction test in a new section at the end of the results to demonstrate that our assumption of no bias correction is not the driver of the results we find (section 3.4, P.15). We have also added some text to the methods to justify not including a bias correction in our main approach (section 2.2, P.6).

*2.2 CO2 fields for the proxy method*

*It would be illustrative to also show a comparison between PR1, PR2, and a third case where the proxy retrievals are constructed using the full-physics XCO2 retrieval:*

*XCH4 proxy =XCH4 no−scatter/XCO2 no−scatter×XCO2 full−physics*

*This could be included in Figure 2, I don't think it's necessary for the authors to perform an additional inversion with this retrieval. It would, however, be nice to show a retrieval that is independent of modeled CO2 fields.*

Whilst we understand the reviewer's viewpoint and agree it would be nice to have a proxy product independent of model CO2, in reality this isn't really directly comparable to the proxy products we use. The advantage of the proxy XCH4 product is that the spatial coverage is much greater than the full physics product due to the need to be less strict on the filtering of the data due to clouds and aerosols in the atmosphere. This is the reason for choosing to use the proxy XCH4 data as opposed to the full physics XCH4 retrieval, where the data coverage over tropical Africa is sparse. The retrieval of XCO2 is obviously beset by the same issues. As such a proxy XCH4 product based on a full physics XCO2 retrieval would lose the key coverage benefit of the proxy product and would not be directly comparable to the PR1 and PR2 data used in this work.

For the same reason we have chosen not to use a full physics XCH4 data product in this work as our initial tests showed there are not sufficient data within the full-physics product to constrain the CH4 fluxes on the scales that we do in the paper using the proxy XCH4 product. As such, the inversion result using full-physics retrievals do not differ markedly from the prior for large parts

of the domain, which is the subject of a manuscript in progress that studies pan-tropical changes in CH4 emissions using GOSAT data.

*2.3 Number of MCMC samples*

*Page 7, Line 10 mentions that the a posteriori distributions each have 2,000 samples. However, aren't there 3,697 basis functions? Does that mean there are less than one sample per basis function? Do these posterior distributions really sample the full space?*

There are 2000 samples that are retained to form the posterior distribution of each parameter (I.e. each a posteriori parameter distribution has 2000 individual values that approximate its PDF). The 2000 values that are stored are a sub-set of the 20,000 iterations that are run in the inversion (the Markov chain is thinned so only every 10th iteration is stored). Each of the 3697 state vector elements are updated at each iteration. This has the joint effect of reducing storage space (2000 instead of 20,000 posteriori samples) and reducing correlation between stored samples if a proposal undergoes a number of consecutive rejections.

We have clarified this in the text with the following statement:

"Updates were proposed to each element of x every second iteration. At iterations in between, updates were proposed to all the theta values."

*2.4 Mention 2010 LST anomaly discussion in the abstract*

*I think it's important that the authors mention that 2010 may be an anomalously low year for wetland emissions in the Sudd (based on their LST analysis). This paper will likely get quite a bit of attention because a number of groups are looking for trends in wetland emissions. The authors do an excellent job of discussing the nuances of their trends in the main text and conclusions, but I think there should be a short (less than one sentence) mention of this possible anomaly in 2010.*

We agree that it is important to mention the caveats in the trend in emissions we have found and have included the following text in the abstract so that this is highlighted up front:

"Using satellite land surface temperature anomalies and altimetry data we find this increase in CH4 emissions is consistent with an increase in wetland extent due to increased inflow from the White Nile, although the data indicate that the Sudd was anomalously dry at the start of our inversion period."

*2.5 Wolfe et al.*

*The authors should mention Wolfe et al., PNAS (2018): "Mapping hydroxyl variability throughout the global remote troposphere via synthesis of airborne and satellite*

*formaldehyde observations" and consider using this data to help constrain their OH fields in the future (would be beyond the scope here).*

We thank the reviewer for bringing the Wolfe et al. (2018) study to our attention. Ultimately it would be insightful to incorporate multiple complementary datasets into an inversion framework in order to constrain separate production and loss processes. We have expanded the discussion section to make this point although, as the reviewer says, it is clearly beyond the scope of this work. We have added the following lines to the text:

"Incorporating additional information from complementary measurement data into the inversion system could further help to reduce uncertainties in the CH4 budget. Examples of potentially useful space-based measurements include formaldehyde to constrain the temporal and spatial variability of the OH radical sink (Wolfe et al. 2018). and GRACE data to help constrain the temporal variability of wetland emissions."

**3 Specific comments:**

*Page 2, Line 23: Suggest replacing mathematically with mechanistically.*

Agreed and replaced.

*Page 3, Line 11: Suggest replacing strong signatures with a synonymous phrase. I typically associate "signatures" with isotopic source signatures, which are also discussed in the manuscript.*

To avoid confusion we have replaced this line with the following text:

"…emissions from seasonal fires …"

*Page 5, Lines 8–12: This paragraph seems out of place, I feel like it should go before the discussion of CO2 fields. Right now it goes –*

*>GOSAT details –>proxy retrievals–>CO2fields –>impact of CO2 fields –>back to GOSAT details.*

To ensure a better flow we have moved this paragraph earlier as suggested.

*Page 5, Line 31: I think you should move the Feng et al. (2017) citation to the beginning of Line 30. It initially sounded like you were building some Kriged global concentration fields*

*from the NOAA data because of this text ". . . coarser global run that was fitted to in situ data. . . "*

Agreed, we have moved the citation.

*Page 6, Line 4: Suggest replacing "prior" with " a priori" for consistency with the rest of the text.*

We have changed the text as suggested to ensure consistency.

Reviewer 2

**General comments**

> *Lunt et al. constrain methane emissions in tropical Africa with GOSAT XCH data to retrieve monthly CH4 emissions in the framework of a Bayesian inversion. A very useful sensitivity test (to the CO2 fields used in the XCH) is included, which makes it possible to provide uncertainty ranges for the estimates of emissions and comment on the significance of the results. With satellite XCH4 data from 2010 to 2016, they are able to study seasonal cycles as well as the 7-year trend. Using various other satellite data (land surface temperature anomaly, altimetry), they are able to suggest links between their findings regarding CH4 emissions and the sources contributing to the variations of these emissions, which are mainly wetlands in this area. This study provides estimates for CH4 emissions and their variations at various spatial and temporal scales in an area, tropical Africa, which is both a key-region for methane emissions and a region where these emissions are very uncertain. The authors have been able to exploit not only satellite data of XCH4 to assimilate in their inversion framework, which already provides rich insights regarding methane emissions, but also to proceed further in the investigation of the possible drivers of the variations of these emissions, making use of different satellite data sets. The manuscript is clearly written and well-structured, the results are very clearly presented and discussed. For all these reasons, I recommend publication of this manuscript in ACP after minor corrections (most corrections for Section 2.3).*

We thank Reviewer 2 for their assessment of the work and address their specific comments below.

**Specific comments**

> *p.2 l.31: what about wild animals for enteric fermentation?*

Based on available literature (e.g. Crutzen et al 1986) we assume wild animals in Africa to mean gazelles and wildebeest. Crutzen et al (1986) estimated 1-5 Tg yr$^{-1}$ for wild ruminants across all tropical regions. Perez-Barberia (2017) estimate 1.1-2.7 Tg yr$^{-1}$ for all wild ruminants globally. As such the source is likely to be small when compared to agriculture and wetlands, and of a size comparable perhaps to termite methane production, although uncertainties on both these source types are large.

We have rephrased this sentence to say:

"The African continent contains significant microbial methane sources from wetlands and agricultural enteric fermentation, as well as smaller microbial sources from termites and wild ruminants (Crutzen, 1986; Sanderson, 1996)."

*p. 2 l.33: indicate what the range corresponds to (one-sigma, 95%, the full range of the ensemble) so that the reader can compare to the other ranges in the paper.*

We thank the reviewer for pointing out the lack of a definition for this range. The range represents the 5th - 95th percentile of the ensemble as defined in Bloom et al (2017). We have added this information to the text.

*p.4 l.4: explanation for the year 2016?*

We can only speculate on the cause of the difference but the timing of 2010 and 2015-2016 suggests El Nino related differences. A recent study has indicated a larger tropical African carbon source using satellite data than is estimated from in situ data (Palmer 2019). This difference would seem to explain the difference between proxy $CH_4$ fields generated from in situ and satellite derived $CO_2$ fields.

In response to this comment we have added the following text:

"The larger PR2 $XCH_4$ levels in 2016 are a result of a larger net $CO_2$ source being inferred from tropical Africa when using satellite data as opposed to *in situ* data (Palmer et al 2019)."

*p.4 l.20: "as opposed to XCO2": I don't understand the idea implied here.*

We acknowledge that this phrase causes some unnecessary confusion and can be dispensed with. We have rephrased this sentence to say:

"In using the ratio to determine $XCH_4$, we assume that atmospheric $CO_2$ varies much less than $CH_4$."

*p.6, Section 2.3: I think this Section is a bit confusing as it seems to be trying to explain a methodology but not in details so that some useful elements are missing for the reader and others are presented in too general a way. Some particular remarks below. Could you try re-writing the Section with a kind of clear hierarchy in the important points for understanding the study described in the paper and the general points which are detailed in the references?*

We thank the reviewer for highlighting the confusing nature of this section and have followed the reviewer's suggestion in attempting to rewrite it in a way that is easier to follow.

Our intention was to provide a brief summary of the method that is described in much greater detail in Ganesan et al (2014) and Lunt et al (2016). However, from the reviewer's comments we see that this may have led to the feeling that we were trying to explain the method in this work, albeit inadequately. To avoid this situation, we have referred the reader to the references for the details of the inversion approach and its benefits. In this rewritten section we attempt to make clear we are briefly summarizing the method by means of introduction before focusing on the specifics of this work, and how the hierarchical approach is set-up.

We have restructured this section to follow the structure below:

A) Brief overview of the general hierarchical Bayesian format

B) Description of the a priori state vector in this work

C) Description of the a priori parameters vector.

D) Description of the sensitivity matrix formulation

E) Description of the MCMC setup.

We hope that this is easier to follow as a result.

Please see the marked up new version of the text and the response to specific comments below for further changes.

*p.6 l.28: "reducing the impact of underlying assumptions": I think it is a bit optimistic to put it like this. The assumptions are still there but they are not made on the same variables/parameters. For instance, the assumption that the errors can be specified as PDFs and, then, the user's choice of one given type and form (be it Gaussian, Poisson or another).*

We agree that investigator assumptions must always be made somewhere and perhaps it might be better phrased as saying the assumptions are one extra step removed from the posterior distribution. I.e. The assumptions are made on the hyper-parameters (i.e. the uncertainty in the uncertainty) and not on the uncertainty parameters themselves. Given the restructuring that we have done to this section in response to the previous point, this line no longer appears in the text.

*p.6 l.30: "uncertainties in the a posteriori distribution are more representative of uncertainties in the inversion system": more representative than what?*

We thank the reviewer for highlighting the phrasing of this sentence and we have rewritten it to say:

"Equation 3 indicates that uncertainties in theta propagate through to the a posteriori distribution of x, which is one of the benefits of this approach given the a priori and model uncertainties may not be well known."

*p.6 l.31: "any form of error variances can be used": do you mean covariances? "Any form" is misleading: they still need to be PDFs, probably ones defined by only a few parameters.*

Indeed, we were attempting to make the point that there is flexibility in the form of PDF that is chosen, and used "any form" within the context of PDFs. We have rephrased this to say:

"In addition, the choice of a priori or measurement PDF is open to the investigator, which allows flexibility in cases where, for example, the PDF should be defined only on the positive axis."

*p.7 l.12: "all parameters in turn": does it mean one parameter per iteration?*

For the avoidance of doubt, we meant that all state vector elements are updated at each iteration. We have rewritten this line to make this point clearer. The line now reads:

"Updates were proposed to all elements of x every second iteration. At iterations in between, updates were proposed to all theta values."

*p.7 l.13: which are the hyper parameters here?*

We realise we have erroneously referred to parameter updates as hyper-parameter updates. To be clearer we define x as the state vector, $\sigma_x$ and $\sigma_y$ as the parameters and the uncertainty in $\sigma_y$ as the hyper-parameters. We have corrected any incorrect specification throughout the text.

*p.7 l.27: what is the type and form of the posterior distribution?*

The posterior distribution is undefined, since it is formed from some combination of Gaussian (likelihood function) and log-normal (prior distribution) distributions, hence the need for a MCMC approach to approximate its form.

In reality, as the violin plots in the new Figure 4 show (see response to p8. L17, fig 3 further down), the form is not too dissimilar from Gaussian.

*p.7 l.20: reference for the measurement uncertainty? Same in both XCH4?*

GOSAT retrieval uncertainties are from the dataset of Parker (2015) and are the same for both PR1 and PR2. We have added this information to the text.

*p.7 l.23: can you comment a bit on the validity of the "uncorrelated" assumption?*

The primary governor of this decision to assume diagonal observation error matrices was the computational cost, however we accept that this assumption of uncorrelated errors is likely to lead to an underestimate of the posterior uncertainties compared to the case where error correlations are included, since too much weight may be given to individual observations.

Correlated measurement uncertainties can, in effect, reduce the number of independent measurements. However, the decision about the scales over which this correlation should apply are difficult to quantify and usually subjective. Chevalier (2007) showed that using error inflation could partly compensate for neglecting the contribution of error correlations, and avoid over-constraining the a posteriori emissions. In our inversions we include both a measurement uncertainty, which is part of the data product from Parker et al (2015), and a second term that is allowed to vary within the inversion and partly acts to inflate the error. We set a minimum value of 5 ppb for this term and constrain it within the inversion. The a posteriori mean value across all timesteps for this additional model uncertainty term is 8 ppb, of a similar magnitude to the 9 ppb mean measurement uncertainty, and leads to an increase in the error term. We note that Turner et al (2015) suggested that the uncertainties of Parker et al (2015) were already a conservative estimate of uncertainty. In addition the mean minimum distance between GOSAT retrievals from the UoL proxy dataset is around 225 km, a distance over which it has been suggested in regional CH4 inversion studies measurements may be largely uncorrelated (Ganesan et al, 2017).

In response to this comment we have expanded on the validity of this assumption in the text as follows:

"Due to the density of data, and to make the inversions more computationally tractable we assumed measurement errors were spatially and temporally uncorrelated. Although this might result in an underestimate of the posterior uncertainties, one approach to form a diagonal approximation of a full covariance matrix is through error inflation (Chevalier 2007).  The inclusion of a model uncertainty term in our inversions, in addition to the measurement uncertainty, partly acts to inflate the error. This additional error term is allowed to vary in the inversion, and constrained by the data, but is of a similar magnitude to the measurement uncertainty, with a mean a posteriori value of 8 ppb, leading to an increase in the error term. In addition, Turner et al (2015) suggested that the measurement uncertainties of Parker et al (2011) were already a conservative estimate of uncertainty, further adding to the observation error inflation."

*p.8 l.11: you should also report the range for the Saunois et al. figure, so that the following conclusion (l. 15-16) that your results are consistent with it is actually supported. You can also make use of Saunois et al. 2017: https://doi.org/10.5194/acp-17-11135-2017*

Agreed. We have added the range of the Saunois estimates in addition to the mean value that was originally included.

*p.8, l.17: Fig 3: it is difficult to interpret the fact that the prior is outside the posterior 95% uncertainty range without any information on the types and forms of the PDFs. Maybe something like Fig 4 of http://dx.doi.org/10.1016/j.spasta.2016.06.005 could be useful?*

We agree that something like the suggested plot would be nice to show as it conveys a lot of information in a single plot. However, it is difficult to apply it in this case, due to the irregular nature of the state vector making it hard to plot the violin plots on a regular grid in a manner that would also convey the geographical location of each of the state elements. We could plot the data as a standalone violin plot but we are conscious of the fact that a reader's geography may need to be guided by a map.

Therefore, we have kept the existing panel c) but added an additional figure containing a violin plot for each of the African regions of the state vector. The regions run from South to North up the y-axis, with the violins showing the a posteriori distributions in Tg yr$^{-1}$, for both the PR1 and PR2 inversions. Red crosses show the a priori mean for each region. We hope it is clear to see from this plot the areas where both posteriors are much greater than the prior mean (e.g. Sudd, Ethiopia, Mozambique, Chad) and those that are much less (Cuvette Centrale and N. DRC).

We have added some additional text that refers to this figure and highlights the regions where the a posteriori distribution is much larger or smaller than the a priori mean.

"This is further highlighted in Fig. 4, which shows the multi-year average a posteriori distributions from both PR1 and PR2 inversions and the a priori mean for each of the basis function regions. The a priori mean value lies outside the 95 percentile range for many of the distributions over the East African regions."

*p.9, Section 3.1: make use of Saunois et al. 2017: they discuss the variability over 2000-2012 so that there are only three years common with your period but you could put your trends in perspective.*

We have added some additional text which makes reference to Saunois et al (2017) in the discussion section (Section 4), where we attempt to put our work in the context of other studies.

*p.9 l.13: "although there was substantial inter-annual variability": I don't understand the logical link between the inter-annual variability (which can go either way from one year to the other) and the trend.*

Indeed, we did not intend to intimate a link between inter-annual variability and the trend. We meant that the inter-annual variability might mask any underlying trends over the inversion period.

We have rephrased this statement to say:

"We found no other consistent trends regional trends between the PR1 and PR2 inversions over the inversion period, although there was substantial inter-annual variability which might mask any underlying trends, particularly over West Africa…"

*p.9 l.22: why are the glint data more sensitive to the boundary conditions?*

The glint data may be more sensitive to the boundaries by virtue of being closer to the boundaries of the domain, and thus any changes to the boundary scaling has a greater impact on the fit to the data. We have added this statement to the text:

"(which are more sensitive to the scale factors applied to the domain boundaries due to their proximity to the edge of the domain)"

*p.11 l. 4-6: indicate whether the link between the variations of LST anomalies and wetland extent variations is a (reasonable) assumption or a proven proxy link. Maybe with a reference?*

LST anomaly data has been used extensively in the meteorological literature as a proxy for soil moisture. When the soil is wet more energy will go to latent heat flux as opposed to sensible heat flux leading to a decrease in the LST. For example, Taylor (2015) compared LST anomalies to independent soil moisture anomalies from the advanced scatterometer (ASCAT) instrument to confirm that LST anomalies are related to soil moisture anomalies through the surface energy balance response. Cammalleri and Vogt (2015) showed that LST can be used directly as a proxy of soil moisture with the best results over areas characterized by water-limited conditions. In response to this comment we have edited these lines to attempt to make this point more explicitly:

"The use of satellite LST data as a proxy for soil moisture is well-established (e.g. Cammalleri and Vogt 2015, Folwell 2016, Gallego-Elvira 2016). In dry conditions, evaporation of water from the soil is restricted, the ratio of latent heat to sensible heat flux decreases and thus the surface temperature increases (Byrne, 1979; Cammalleri and Vogt, 2015). As such, areas of elevated soil

moisture exhibit cooler surface temperatures than dry soil and hence lower LST. The LST anomaly from the climatological monthly mean LST …"

*p.13 l.4: "r2 values of 0.2-0.8": 0.2 does not seem to be such a strong correlation. Do you have criteria for the significance of this?*

We thank the reviewer for highlighting this, and we have expanded on the correlation coefficients and added them to each of the individual panels of Fig. 7 to be clearer about which regions show a reasonable correlation between emissions and LWE anomalies. The 0.2 figure is from the Angola/Zambia region which has a significant biomass burning component to emissions which would appear to lessen the correlation to LWE anomaly. The largest correlation coefficients should be 0.7 (not 0.8 as was written in error) for the Chad, Sudan, South Sudan, Madagascar and Niger Basin regions. Some of the regions exhibit a smaller secondary emissions peak that reduces the correlation with LWE anomaly, but the correlation between LWE anomaly and the major seasonal CH4 emissions peak remains dominant. The text now reads:

"Chad, Sudan, South Sudan, Madagascar and the Niger Basin in particular show the strongest correlations with $r^2$ values of 0.7 as shown in Fig. 8."

*p.15. l.1 seq: make use of Saunois et al. 2017.*

We attempted to use references that covered global emissions over a period similar to the one in this work. However, we are happy to also make use of the Saunois reference if it helps to put the results in context. We have added the following lines to the text:

"Although Saunois et al (2017) found emissions from the tropics to have increased by 18 (13-24) Tg yr$^{-1}$ between the periods 2002-2006 and 2008-2012, relatively little of this was ascribed to Africa. This is qualitatively consistent with our hypothesis that the increase in emissions from East African regions due to increased water levels of the East African lakes was limited to the period between 2010 and 2016."

**Technical corrections**

*fig 1 (a): what is the background of the map: climatological vegetation cover?*

The background is a shaded relief map from natural earth https://www.naturalearthdata.com. We have added this information to the figure caption.

*p.3 l.19: "Gravity Recovery and Climate Experiment (GRACE,"->And*

Corrected.

*p.3 l.20: "liquid water equivalent height (LWE) height anomaly retrievals" -> delete first height*

Corrected.

*p.4 l.31: "different to" -> different from?*

 Corrected.

*p.7 l.28: "clearly show" -> clearly shows*

Corrected

*p.8 l.17 seq.: there are a lot of numbers in the two paragraphs: would it be possible to make a table? E.g. with columns prior, PR1 posterior, PR2 posterior. Same remark for p.9 l.2 seq., p. 10 first paragraph*

Whilst we appreciate it may not be to everyone's liking, we have included several numbers in the text as we feel this it is important to quantify our results. We have also endeavoured where possible to present these graphically through the figures. We hope the inclusion of the additional violin plot (Figure 4) has helped to convey the results more clearly and feel that this minimizes the need for any additional tables.

*p.8 l.33: "that" -> than (smaller l. 32)*

Corrected.

*p.9 l.12: "reigon" -> region*

Corrected.

*p.10 l.18: "represents" -> represent*

Corrected.

*p.11 l.4: "anomalies from" -> "anomalies of"?*

Corrected.

*p.11 l.30: missing )*

Corrected.

*p. 15 l.32: "changes" -> change*

Corrected.

[revised manuscript text omitted]